# An evidence-based guidance framework for neural network system diagrams

**Guy Marshall**[1]*, **André Freitas**[1,2], **Caroline Jay**[1]

**1** Department of Computer Science, University of Manchester, Manchester, United Kingdom, **2** IDIAP, Martigny, Switzerland

* guy@fuza.co.uk

**Data availability statement:** All supporting files are available in figshare (DOIs: Interview transcripts: 10.6084/m9.figshare.12765596.v1, Evaluation data: 10.6084/m9.figshare.22698526.v1, Citation data: 10.6084/m9.figshare.14812959).

**Funding:** GM: PhD stipend awarded by University of Manchester Department of Computer Science. The funders had no role in study design, data collection and analysis, decision to publish, or preparation of the manuscript.

## Abstract

Accurate communication of research is essential. We present the first evidence-based framework for formatting neural network architecture diagrams within scholarly publications. Neural networks are a prevalent and important machine learning component, and their application is leading to significant scientific progress in many domains. Diagrams are key to their communication, appearing in almost all papers describing novel systems. However, there are currently no established, evidenced-based conventions describing how they should be presented. We study the use of neural network system diagrams through interviews, card sorting, and qualitative feedback structured around ecologically-derived example diagrams. We find that diagrams in scholarly publications can be difficult to interpret due to ambiguity and variance in their presentation, and that there is a high diversity of usage, perception and preference in both the creation and interpretation of diagrams.

We examine the results in the context of existing design, information visualisation, and user experience guidelines and use this foundation to derive a framework for formatting diagrams, which is evaluated through an experimental study, and a comprehensive "corpus-based" approach examining properties of published diagrams in top neural network venues. The studies demonstrate that 1) both the usability and utility of the framework are high and 2) papers containing diagrams that conform to the guidelines receive more citations than those containing diagrams that violate them.

## 1 Introduction

This paper aims to identify an approach to measurably improve scholarly neural network diagrams, and does so by proposing and evaluating a framework of diagramming guidelines. Neural networks are often used in Artificial Intelligence (AI) systems. Two important application domains are Natural Language Processing (NLP) and Computer Vision (CV), specialising in the creation of systems to perform tasks involving language or images respectively. In addition to the core areas of classification and pattern prediction, neural network systems have been successfully applied in complex domains such as autonomous driving, language translation, or automated question answering.

**Competing interests:** The authors have declared that no competing interests exist.

Increasingly complex and niche application areas are being identified, and systems built to address these problems. SemEval, an annual semantic evaluation workshop, has different tasks each year. In 2020 the tasks included Memotion Analysis (the analysis of internet memes), Detection of Propaganda Techniques in News Articles, and Multilingual Offensive Language Identification in Social Media [1]. Neural network systems have demonstrated the potential to address a wide range of modern issues. In some cases, neural network systems have been created which outperform humans by a considerable margin, such as recently found in biology, in an image classification task, where a neural network's 90% accuracy significantly outperforms the 50% human expert accuracy [2]. With such a wide range of useful application areas, and with such demonstrable potential advancement, there is a huge amount of scholarly activity related to neural networks.

As in other disciplines, scholars communicating advances to their community do so through journal and conference papers, which often include a system diagram. Interpretation of these diagrams is an important part of scholarly communication about neural network systems. An example diagram is shown in Fig 1.

Incorrect interpretation of these diagrams has the potential to cause misunderstandings about the system design, leading to an incorrect understanding of the scientific advancement by other researchers. For scientists and software engineers applying the research in their application areas, there is the risk of wasting time in applying unsuitable techniques, again through a lack of proper understanding of the system. For these reasons, accuracy, clarity, and overall effectiveness of system diagrams is important.

We use an interview study in order to capture rich, individual feedback about diagrams. We explore a broad range of topics about the use of diagrams, and uncover preferences and communication issues, in order to generate requirements for a diagram improvement framework.

The framework is then evaluated with ten AI researchers, in a multi-stage mixed-methods study. In this empirical study, participants edit their own diagrams according to the framework, and provide quantitative and qualitative feedback on the diagrams of others, both before and after exposure to the framework.

To provide a different evaluation of the framework we additionally employ a corpus-based approach, which explores how framework compliance relates to citation counts of diagrams found at ACL 2017 [4]. This analysis is based on previously published research data, considering this data from an evaluation perspective.

In summary, this work (i) identifies diagrams as important in scholarly communication about NN systems and (ii) establishes and evaluates a framework for improved NN system diagrams.

## 1.1 The rapid growth of AI systems research

Fig 2 shows main track long paper submissions to ACL (Natural Language Processing) and CVPR (Computer Vision) conferences from 2009 to 2021. Note the rapid increase in submissions since 2017. These conferences have the highest h5-index and highest attendee numbers in their domains, and have similar (approximately 25%) acceptance rates. From 2017 to 2021, there were 180% and 154% increase in submissions for ACL and CVPR respectively. By contrast, SIGCHI, a similarly top h5-index venue in the domain of Human-Computer Interaction, had a 19% increase in submissions over the same period [11–13]. The large number of submissions comes with administrative issues for organisers, and also for researchers in remaining current with the field. For example, in 2021, ACL moved to a "rolling review"

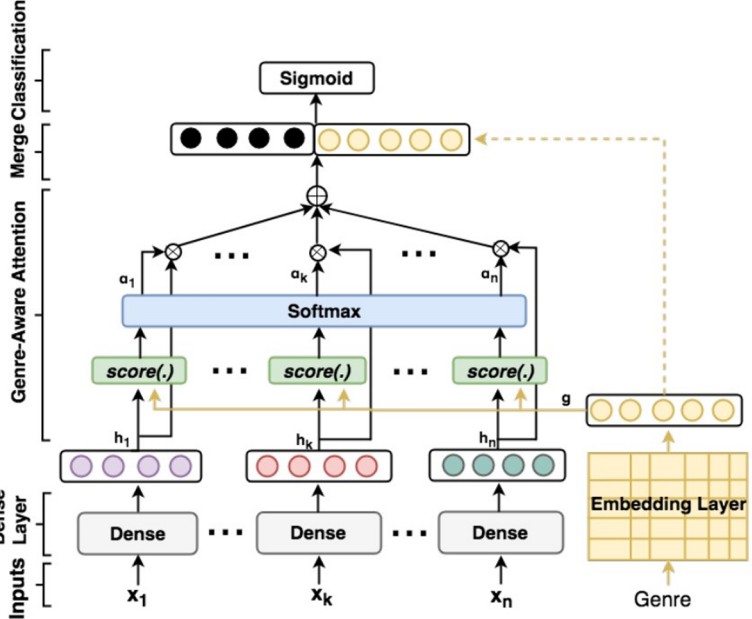

**Fig 1. An example scholarly neural network system diagram, from Maharjan et al.[3].**

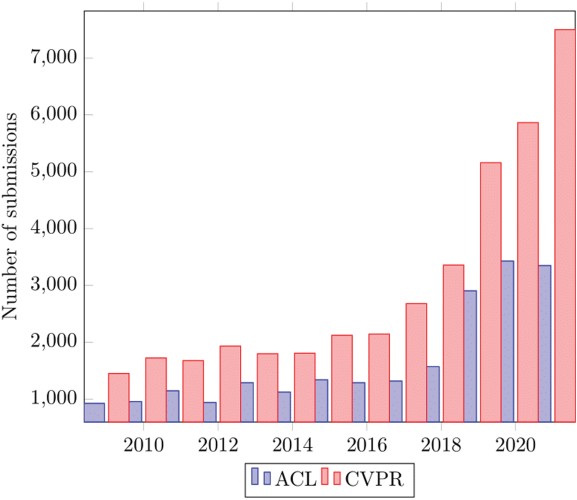

**Fig 2. ACL [5–7] and CVPR [8–10] full paper submissions.**

process, in part to spread the reviewing load over a wider time period https://aclrollingreview.org/. The fast pace of research increases the importance of effective communication.

From a scholarly communication perspective, AI research artefacts can be considered as consisting of journals, conference proceedings, web-pages, code, pre-prints and peer reviews. Perhaps in part due to the fast pace of development the field, conferences are particularly prestigious in Computer Science [14], and are of a constrained format and scope. For these reasons analysis and discussion is focused on conference proceedings.

Having effective scholarly communication about AI systems is crucial for communication efficiency, allowing scholars to stay at the cutting edge despite the increasing volume of publications. Effective communication also supports accurate interpretation, critical assessment and building upon published work.

## 1.2 Neural network systems

Neural network systems are usually designed and trained to perform a specific task, such as classifying images or predicting the next word in a sentence. A neural network system can be considered to encompass the entire software system, rather than a distinct neural component in isolation. This scope corresponds well to the content commonly included in diagrams in scholarly publications.

A neural network takes an input (such as text or images), and then processes this via a series of *layers*, to arrive at an output (classification/prediction). Within each layer are a number of *nodes* that hold information and transmit outputs to nodes in other layers. Specific mathematical functions or operations are also used in these systems, such as sigmoid, concatenate, softmax, max pooling, and loss. Hyperparameters are parameters used to control the learning process, such as the learning rate, and are often tuned for each system implementation. The *system architecture* describes the way in which the components are arranged. Different architectures are used for different types of activities. For example Convolutional Neural Networks (CNNs) are commonly used for processing images. Long Short Term Memory networks (LSTMs), a type of Recurrent Neural Network (RNN) which are designed for processing sequences, are often used for text.

Neural networks "learn" a function, but have to be trained to do so. Training consists of providing inputs and expected outputs, allowing the system to develop an understanding of how an input should be interpreted. The system is then tested with unseen inputs, to measure whether it is able to handle these correctly and generalise to new cases. A more detailed introduction to the field is provided by Goodfellow et al. [15].

## 1.3 Diagrams in communicating neural network systems

Diagrams are a useful way of representing general systems. Peirce, an American philosopher and semiotician, defines diagrams as "icons of relation" [16]. In Cybernetics, a system can be defined as "an integral set of elements in the aggregate of their relations and connections" [17]. The shared emphasis on relations suggest that diagrams may be a suitable and useful representation for systems.

In practice, diagrams are a prevalent medium for communicating neural network systems. Examining neural network system diagrams found in conference proceedings, there are few conventions. There is variety at a structural level, in terms of what is represented (be it inclusion of an example, data shapes, processing steps, or class names), the level of granularity, and how it is represented (such as blocks, graphical icons, natural language, or mathematical notation). At a lower level there is variety in how fundamental elements such as vectors are represented as graphical components, sometimes even within the same diagram. This contrasts with terms in text, equations, pseudocode, and code, which are predominantly formalised and consistent.

## 1.4 Diagrams in scholarly NN system publications

The use of NNs in scholarly publications could be considered as models, architectures, or systems. The term "system" is used throughout to avoid confusion with the design of neurons

themselves (sometimes termed NN "architectures"), and to reflect that the diagrams in practice may include elements beyond the neural "model" itself, such as the application context.

The present situation of diagramming of NN systems at scholarly venues is varied, with few conventions. This is in terms of content, visual encoding, tools and usage. Fig 3 shows a selection of diagrams from an ACL 2017 diagram corpus created as part of this research program [18], and gives a quick visual indication of this heterogeneity that motivates this investigation.

The top left diagram uses coloured circles to represent tensors with mathematical labels, the bottom uses mathematical notation within boxes, as does the top right (though it also uses circles as tensor output). The top right and bottom appear visually similar, but the bottom includes an example input and output, uses straight lines only, has overall data flow right to top, and describes multiple models within one diagram. These content features all differ to the top right. Fundamental differences in diagrams can also be seen with diagrams which are at a higher level than tensors, or diagrams using modules or sub-figures to operate at varying abstraction levels.

This multifaceted heterogeneity is worthy of study because in other scholarly domains formal standards and "conventional" practices are often established and employed. The nature of NN systems, being conceptually-non-linear, multi-component, high-dimensional data systems, makes communication about NN systems challenging and diagrams are a medium used for this within scholarly publications. The heterogeneity suggests there may be opportunity for improvement in diagramming practices.

Research to date has utilised VisDNA, a grammar of graphics, to analyse scholarly NN diagrams [22], demonstrating the heterogeneity of the domain by using all visual encoding principles. There have been various attempts to classify scholarly NN system diagrams [23–27]. Of particular note is Net2Vis, due to the supporting qualitative research.

## 1.5 Net2Vis research

In creating Net2Vis, Bäuerle et al. [25] state requirements for CNN visualisations, such as requiring that model properties and layer properties be visualised. A qualitative evaluation of the Net2Vis visualisations was done with 7 experienced Machine Learning (ML) researchers, gathering qualitative data in a survey format. They used Munzner's nested evaluation model [28], a visualization design and validation framework , and "assessed the need for such automatic visualizations (Q1, Q5), analyzing the threat of targeting a wrong problem. We also investigated why 3D visualizations are so common (Q3), and asked about our visualization design (Q2, Q4) to evaluate the abstraction and encoding technique, which are the second and third possible pitfalls." [25]

To summarise their results, figure creation was said by all expert participants to take "too much time". They found that 3D visualisations were used by three experts only to convey that the data was three dimensional. These experts also noted that this made the diagram more complex.

Following amendments to their system based on the qualitative expert feedback, they evaluated with 10 less experienced ML researchers. They compared their visualisation with TensorBoard and the original visualisation, for several well known architectures. Participants were asked eight quantitative questions about the architecture: "How many convolutional layers does this architecture contain?, What is the maximal feature depth for the convolutional part?, What is the minimal spatial resolution of the convolutional part?, What are the input dimensions for this network?, What are/is the output dimension(s) of this network?, How many times does downsampling happen in this network?, How many steps are performed

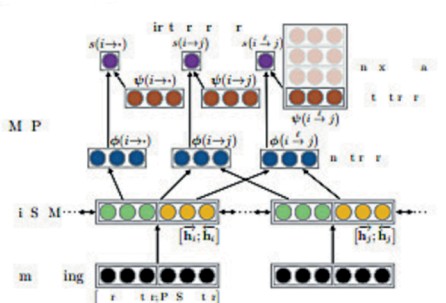

**Figure 3:** Illustration of the architecture of the basic model. $i$ and $j$ denote the indices of tokens in the given sentence. The figure depicts single-layer BiLSTM and MLPs, while in practice we use two layers for both.

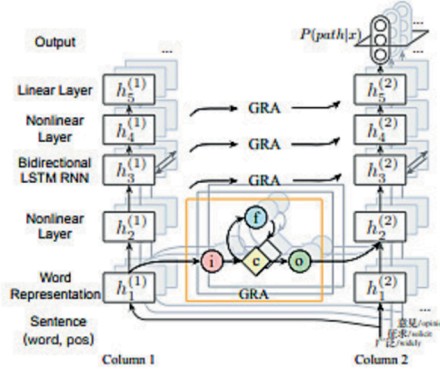

**Figure 3:** Each column is a stacked bidirectional LSTM RNN model. Two columns are connected by GRAs. There are three gates in each GRA: $g_i$, $g_f$, and $g_o$. The input gate $g_i$ and the forget gate $g_f$ can also be coupled as one uniform gate, that is $g_i = 1 - g_f$.

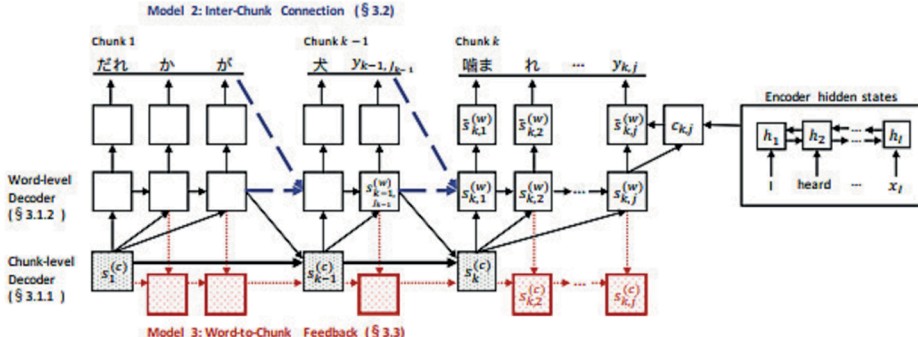

**Figure 4:** Proposed model: NMT with chunk-based decoder. A chunk-level decoder generates a chunk representation for each chunk while a word-level decoder uses the representation to predict each word. The solid lines in the figure illustrate Model 1. The dashed blue arrows in the word-level decoder denote the connections added in Model 2. The dotted red arrows in the chunk-level decoder denote the feedback states added in Model 3; the connections in the thick black arrows are replaced with the dotted red arrows.

**Fig 3. Three example NN system diagrams from ACL 2017, with captions from original papers, licensed under CC BY 4.0.** Clockwise from top left: [19–21] (IDs numbers as in the diagram corpus of [18]: 186, 189, 174).

to increase the feature dimension?, Is this Architecture 'Fully Convolutional'?" [25]. These questions were somewhat grounded on the aforementioned qualitative survey, and have an engineering focus. They measured the accuracy of answers, finding their visualisations had a higher average accuracy and lower variance in accuracy than TensorBoard. They did not find a significant difference between their Net2Vis visualisation and the original papers' visualisation. Note that, as with the present method, they evaluate and examine the diagrams out of the paper's context.

They also evaluated their visualisation system with 16 visual designers, using a system usability scale questionnaire, and reported "excellent" usability. The participants had never

created a CNN architecture diagram before, but had recently received training on deep learning concepts.

## 1.6 The potential of diagrams

Diagrams can be useful aids for describing, interpreting and reasoning about systems. Commentators have praised diagrams for controlling search space [29], their explicit spatial relation advantage [30] and explanatory value [31]. Stenning and Oberlander [32] state that the power of visual representations is in the omission of information, limiting abstraction to aid "processibility". In a similar vein, Levesque [33] argues for simplicity, that inferential and computational tractability is maintained by minimising the number of cases that must be computed over. Shimojima's work concerning "free rides" [34] includes a number of examples, the core concept of which is that by establishing one relationship in a diagram there is also established a relationship to all objects within the diagram. This leads to claims for diagrams aiding inference and consistency checking, at the expense of potential over-specificity. To summarise these works, there is evidence that diagrams:

- Are external representational support to cognitive processes [35].
- Make topics simpler, leading to reduced search space and fewer cases to be computed over, by including minimal salient information [29].
- Are manipulated in order to profile known information in an optimal fashion [36].
- Make abstract properties and relations cognitively accessible [37].
- Facilitate perceptual "free rides" in inference, making relations evident that might not be obvious in a different representation [34].
- Can be in a public space, therefore enabling collective and temporally distributed forms of thinking [16].

These attributes provide evidence as to why diagrams may be an appropriate medium for scholarly reasoning and communication, including for scholarly NN systems.

## 1.7 Lack of guidance for diagrammatic practices in scholarly communication

Diagrams are seen as important by some scholars. Carberry et al. [38] note that information sometimes resides in figures that cannot be found elsewhere in the text. This suggests that diagrams contain content not available elsewhere, and as such may have an important and unique role when reading and extracting information from a paper. Rowley [39] examines the academic conference presentation as a medium, investigating the different types of imagery used, from photographs to system diagrams. Rowley found that in Physics 52% of slides used in conferences were of images, including diagrams and charts. This highlights the prevalence of diagrams as part of scientific scholarly communication more broadly.

Whilst figures and diagrams may be seen as important by some scholars, and diagrams are certainly prevalent in many domains, discussion of these is limited in popular academic writing guides. Swales and Feak's "Academic Writing for Graduate Students" [40], despite including 11 conceptual diagrams to explain their own work, only gives guidance for the use of charts, not for the use of other figures such as system or conceptual diagrams. In 212 pages, the single mention of figures or diagrams in Murray's "Writing for academic journals" is the rhetorical question "Do you have any figures, diagrams or tables to include?" [41]. Hall's medically-focused "How to write a paper" [42] discusses some specific areas related to diagramming, providing extensive advice for captions, legends and referencing the figure in text,

and advising brevity and minimising duplication for the content of diagrams. Hall deals with figures and illustrations primarily relating to graphs in the "Results" section, and also notes in the "Methods" section that "A diagram can help a lot to describe a complex study design or sequence of interventions" [42] . This is the only reference to system diagrams. No further guidance on content or presentation of diagrams is given. Schimel's "Writing Science" includes limited advice on referencing a chart in the text, and their advice on diagrams and figures extends only to the following comment: "I have always felt that I don't understand something until I can draw a cartoon to explain it. A simple diagram or model - the clearer the picture, the better" [43].

None of the above paper writing guides includes a chapter, section or subsection discussing diagrams. These examples are indicative of the usual level of diagram discussion in highly cited scholarly paper-writing guides. There are exceptions to this brevity. One such relevant domain-specific paper writing guide providing some depth of diagramming advice is "Writing for Computer Science", in which Zobel [44] includes one chapter and two additional subsections about figures. This includes tables, algorithm figures, graphs, and figures in slide presentations. For system diagrams, Zobel suggests making use of sketches, using available diagrammatic languages for the specific domain, and outlines general design considerations such as removing clutter. These guidelines are not evidence-based, and are published without references, making auditing difficult, though the guidelines appear to replicate the advice of Tufte [45]. Zobel [44] notes that:

> Diagrams illustrating system structure often seem to be poor. In too many of these pictures the symbolism is inconsistent: boxes have different meanings in different places, lines represent both control flow and data flow, objects of primary interest are not distinguished from minor components, and so on. Unnecessary elements are included, such as cheesy clip-art or computer components that are irrelevant to the system.

In summary, whilst some scholars note the importance of diagrams for scholarly communication, there is a lack of support for diagramming in scholarly writing guides.

## 1.8 Research questions

The goal of this study is to propose a way of creating measurably better scholarly neural network diagrams. To do this, the study addresses the following research questions, in the domain of neural network systems:

- Why do people create system diagrams for scholarly papers? (Interview)
- How do people create them? (Interview)
- What tasks do system diagrams support for readers of scholarly papers? (Card sorting; Interview)
- What aspects of presentation do people find helpful or confusing? (Example diagrams; Interview)
- What measurable changes does exposure to the proposed framework cause? (Empirical evaluation)
- How do current diagramming practices relate to the proposed framework? (Corpus evaluation)

The research questions are designed to gathering requirements for potential diagrammatic tools and identify avenues for future research. Additionally, this information is useful to researchers in the domain of neural network systems, to inform diagram design.

We find a large variety of opinions, with only slight agreement on preference of example diagrams, task importance, and diagramming tools. We also identify a number of areas causing confusion to readers, such as whether a precise depiction is meaningful, and the omission of expected details from a diagram. We also report that for some readers, scholarly system diagrams provide an overview of the system, allowing them to quickly understand a paper. This highlights the importance and unique role of diagrams in scholarly communication.

## 2 Related work

### 2.1 Why do people create diagrams?

There are good reasons for using diagrams generally, which also apply to system diagrams. Diagrams make abstract properties and relations accessible [37,46]. They are external representations which support cognitive processes [35,47]. Further, in a public setting, they can enable collective or distributed thinking [16]. Diagrams can also be "manipulated in order to profile known information in an optimal fashion" [36]. Cognitive and perceptual benefits of diagrams for handling complexity are well documented, particularly in their ability to limit abstraction and aid "processibility" [48]. Each of these attributes of diagrams has the capability to support research processes.

More specifically, there are benefits in using visual representations to display information. Van Wijk's [49] economic model provides quantification of this value, by adding "cost" to each activity. For example, a useful business information visualisation that reduces employee time taken and is frequently used gives a measurable financial benefit, and the financial cost of initially building and maintaining the visualisation determines the return on investment, and whether the visualisation is good value. From an InfoVis perspective, diagrams make information more useful by removing noise, and improve the accessibility of complex algorithms [50].

In scholarly documents, Tenopir et al. [51] use a survey and a series of user studies to understand readership use cases of figures within scholarly documents and to test prototypes for ProQuest, a digital research library. The prototype was tested in ecological science, and involved extraction of data, including figures, into a metadata page of "disaggregated components". Their conclusions are primarily about researcher activity using these components, noting "emerging opportunities to conduct research into scholarly communications focused on artifacts at finer levels of granularity". From their hands-on study, they identify four main readership uses of figures: (i) "creating new fixed documents", (ii) "creating documents to support performative activities", (iii) "making comparisons between a scientist's own work and the work of other researchers", and (iv) "creating other information forms and objects" [51]. Of relevance to diagrams, Tenopir et al. state that "in-depth indexing is applied to individual tables and figures, which allows searchers to locate information of interest even if the entire article is not on that topic" [51]. Referring to a lack of science about science, they noted more generally that "investigations of scientists' use of journal articles for purposes other than research have been rare" [51].

In education, cognitive benefits of diagrams have been researched in Venn diagrams, tree diagrams and other representations "encouraging thought regarding the whole and its parts" [52]. In a recent meta-study, Guo et al. [53] showed diagrams had a moderate overall positive effect on the comprehension of educational texts. Both research and education require information searching behaviour, so it would be reasonable to expect some elements of these education domain results to also hold in the research domain. However, compared with research tasks which are primarily communicative, education tasks have different, pedagogical, desired outcomes. See Tippett [54] for a systematic review of visual representations

in science education. The substantial differences in user profiles, use cases and representational choices compared with scholarly research lead us to exclude the education domain from further discussion.

## 2.2 How do people create diagrams?

In terms of the process for creating diagrams, cognitive theories are helpful for understanding how people summarise and integrate information. There is a close relation between "how" and "why" diagrams are created, particularly in terms of perceptual and cognitive attributes. As such, work found in Sect 2.1 is also relevant to this research question.

In an interview study conducted retrospectively with building architects, Suwa and Tversky [55] use a protocol analysis to demonstrate the utility of sketches in "crystallizing design ideas".

Conceptual diagramming, using diagrams to support the cognition of concepts, can be considered a closely related domain, if one considers a system architecture to be a conceptualisation of the design. In an interview study of conceptual diagramming, Ma'ayan et al. [56] investigate how people draw diagrams relating to complex concepts, including computer systems, in order to generate requirements for diagramming tools. They focus on what they term "natural diagramming", which refers to the author having a direct relation between their conceptualisation and the diagram, and being able to use the diagram to explore a conceptual space.

## 2.3 What aspects of presentation do people find helpful or confusing?

Different ways of writing things down can lead to vastly different outcomes, both in natural language [57,58] and diagrams [34], particularly mathematical education diagrams [59–61]. Specific graphical components used in diagrams can convey entirely different meanings, aiding or hindering accurate interpretation. Considering diagrams as analogous to cities, and design of diagrams analogous to architectural design, Blackwell notes that the design decisions taken in diagramming make certain kinds of experience more likely or less likely [62]. In mechanical drawings, experiments have shown that the addition of arrows alters a structural diagram to convey functional information [63]. Physics of notation, a diagram analysis framework proposed by Moody [64] which is sometimes used to design new notations [65], includes a category for "semantic transparency", where chosen visual representations automatically suggest their meaning.

## 2.4 What tasks do system diagrams support?

We are not aware of any prior work examining *usage* of scholarly neural network system diagrams. There is a substantial body of empirical systems diagram research for Unified Modeling Language (UML), a diagrammatic language used for software diagrams [66]. The tasks commonly studied in empirical research are for software engineering, rather than software research, and often prioritise error detection [67] or maintenance [68], alongside more generally applicable diagramming topics such as cognitive integration [69] or comprehension [70].

Tasks that researchers ask participants to perform using diagrams are often stated without evidence or discussion, such as the examples given above. In experiments on flow maps with non-specialist users, Koylu and Guo [71] conclude that "[t]he influence of the design on performance and perception depends on the type of the task", suggesting this is a useful research question.

## 2.5 Guideline evaluation methods

**2.5.1 A lack of evaluation of guidelines**   uidelines are evaluated with a range of methods and metrics. These range from empirical to theoretical; covering usability and/or utility, usually for specific tasks. The literature review is not intended to be comprehensive, but maps the territory of guideline evaluation in several domains. I am not aware of any empirical work evaluating software diagram guidelines. Smith [72] notes that

> One significant aspect of our software design standards and guidelines is that they are largely based on expert judgement and accumulated practical experience, rather than on experimental data and quantitative performance measures.

A common practice in Software Engineering is to propose a new diagrammatic or visual notation, and perform a comparative evaluation of the new notation (either against text e.g. [73], or against an existing visual notation e.g. [74]). In evaluating new notations or guidelines, there is necessarily a comparison of two visual representations of the same underlying system to test whether it performs better than the existing notation. The methods and metrics for comparative analysis of two notations relate to (i) a substantial visual and content change (ii) often comparing for a particular task or expressed preference (iii) cannot aim to understand individual changes to the representation. Application of guidelines to an existing diagram can be considered as a different process to the usage of an entirely new notation. As such, related work and established methods for *UI guideline evaluation* is prioritised over *software diagram notation comparison*.

In the domain of software visualisation, evaluation methods have been recently systematically reviewed by Merino et al. [75]. They report that 62% of the proposed software visualization approaches examined did not include a strong evaluation. Those conducting evaluation did so primarily collecting data through questionnaires, interviews or think aloud studies.

In a systematic review of research-derived touchscreen design guidelines for older adults, Nurgalieva et al. [76] found "proposed guidelines and recommendations were validated in only 15% of articles analyzed".

In user interface guidelines, an influential scholarly domain, a variety of empirical evaluation methods and levels of rigour are employed. Many do not include an evaluation. For example, in intelligent television, only one of five prior studies cited by Kunert [77] included an evaluation. The single study with an evaluation did so by conducting a survey of developers (the users of the guidelines) and a usability evaluation on 11 existing applications [78].

**2.5.2 Experimental comparison of different notations**   Gross and Doerr [79] conducted two distinct experiments to compare "event-driven process chain diagrams" (EPC diagrams) and UML Activity Diagrams. Whilst this study is not about guidelines, the method is useful due its empirical evaluation of authorship and readership activities. The first was an engineer perspective, and the second a customer perspective. They attempted to assess both efficiency and effectiveness, through complexity of diagram and correctness of models, respectively.

Their first experiment assessed efficiency through the proxy of complexity and correctness. The study consisted of giving a tutorial and asking students to draw diagrams by hand. They split participants into two groups, and each group took one diagram format as a "treatment", with 60 minutes to complete the task. The complexity was measured by number of elements, with Levene's test and t-test alongside descriptive statistics. Correctness was measured by number of errors, there being a correct answer. The second experiment assessed effectiveness, through the proxy of interpretation correctness. Participants were different to the previous experiment, and had no prior experience. Again they were split into EPC and UML groups, using new diagrams provided by the investigators (i.e. not from the first experiment). This

experiment had two parts, the first answering 14 content-related questions, and the second identifying errors in erroneous diagrams, compared to a textual description. A participant questionnaire captured difficulties and experience. By using a study design engaging both authors and readers, evaluation of the utility of guidelines can be framed as a comparison of two visual notations, making the above studies relevant.

**2.5.3 Studies informing methodological choices** Methodologically, in order to evaluate the performance of guidelines, insight can be gained from a number of studies which consider multiple perspectives. Steering methodological choices is a theoretical semiotic work about measuring diagram quality [80], which advocates measures covering authorship and readership practices, and the gathering of multiple qualitative and quantitative metrics.

Colwell and Petrie's [81] evaluation of web content accessibility (WCA) guidelines utilised two experiments: Experiment 1 was adapting existing html pages using think aloud (P = 12). Experiment 2 was remotely administered to visually impaired people (P = 20) not involved with Experiment 1, and given an evaluation questionnaire based on some example elements created in Experiment 1. As part of that, they had to perform a task to interrogate a table to extract information. Both experiments had unexpected outputs which were transmitted to the "WCA Guidelines Working Group" and led to some changes to the guidelines. The outputs were very domain specific, such as "tables seemed to be more accessible than expected and the alternative text for images less accessible". No quantitative metrics were provided.

Eichelberger and Schmid [82] propose aesthetic guidelines for UML, based on previously published investigations. In a relatively intricate experiment, they evaluated their guidelines with 18 students manipulating 6 example diagrams from different domains. These examples were modified in different ways to violate a guideline. They hypothesised adherence to guideline reduces the number of faults found and reduces the time required, but did not find sufficient information to disprove the null hypothesis for an individual guideline. They tested the null hypothesis with ANOVA. They also asked 2 questions about understandability and modifiability. They conducted a preferences questionnaire which was not reported as it contained no "unexpected results". They concluded that the domain effects had a larger impact than guideline observance. This result informed methodological choices of the present study, which quantitatively assesses the overall guidelines at task-level, while gathering qualitative user feedback on individual guidelines. However, unlike Eichelberger and Schmid, the present method does not quantitatively investigate the impact of individual guidelines on the end-users.

Al-Saʾdi [83] proposes guidelines for the field of Arabic language Jordanian educational UIs, and uses an iterative approach to validate guidelines, refining and verifying with designers and developers. Through the course of this thesis, the series of studies conducted were (i) an interview study, (ii) showing examples to users using think aloud, (iii) iterating the proposed guidelines using the Delphi method. The Delphi method could be considered as an implicit qualitative evaluation by experts. No other evaluation of the guidelines was performed.

Zajonc [84] argues that repeated exposure to a stimulus object enhances attitude towards it. As such, in the present study the exposure order of diagrams is randomised to remove the possible confounding effects of priming.

## 2.6 Scholarly interview studies.

In their interview and prototype study of scholarly information, Pontis et al. [85] identified different attributes, such as experience level and the project's state, influencing researchers' information-seeking behaviour. Pain points, uses and strategies are described through the

information journey. The main conclusion was that better support for researchers in filtering content is important for future scientific information tools. Use of diagrams was not reported.

Research conducted to support the design and evaluation of Net2Vis included interviews with diagram authors. Figure creation was said by all expert participants to take "too much time", supporting comments made by participants in this study. Further, their corpus research highlighted the heterogeneity described by participants of this study, and differs to the Vis-DNA approach by including content as well as visual encoding.

As noted in Sect 1.7, there is a lack of guidance provided about creating and using scholarly diagrams. The present research begins to address an important omission in research into scholarly practices, especially concerning the use of diagrams as a summary, in initial relevance filtering and as an aid to instantiate an example.

## 3 Method

### 3.1 Overview

The scope of the interview study centres on readers of publications at six top Neural Network venues: ACL, NAACL, EMNLP, CVPR, ECCV, and ICCV. This scope was chosen to restrict scope and to ensure a NN specialism. The interviews, including card sorting and graphical stimuli, was conducted by video (6 participants) or in person (6 participants). Analysis took place in NVivo (qualitative thematic analysis) and in Excel and R (quantitative). The results here The output of this study is a set of guidelines.

The scope of the guidelines framework evaluation is also readers of publications at six top Neural Network venues: ACL, NAACL, EMNLP, CVPR, ECCV, and ICCV. The experiment was conducted via email, with analysis in Excel and R. The output of this study is a set of ten before and after diagrams (as .png files), and participants' assessments on the utility and usability of the guidelines framework.

The corpus analysis is scoped to all publications in ACL 2017 only, due to the significant effort required to generate and evaluate the diagram corpus. The corpus-based approach was chosen to evaluate from a different (and broader) perspective the applicability of the guidelines. Analysis took place in R.

Limitations are discussed for each study within their respective methods, specifically in Sects 3.3.2, 3.2.3, 4.9, and 4.11.10.

### 3.2 Interview study setup

We conduct a semi-structured interview study, and including the examination of six example diagrams and a closed card-sorting exercise to identify "useful and not-at-all-useful tasks". University of Manchester Department of Computer Science ethical review board approval was granted for this study (2019-7852-11951). Full interview scripts and transcripts have been made available [86].

**3.2.1 Participants** We recruited 12 participants (Table 1), each reporting having read at least one paper from the top three H-indexed computer vision or natural language processing conferences, in the last 12 months (ACL, NAACL, EMNLP, CVPR, ECCV, or ICCV). Recruitment took place from 14th November 2019 to 12th December 2019. Written consent was collected. All participants were previously known to the research team, though not necessarily the interviewer, and spanned seven academic, academic related, and commercial institutions. The self-reported specialisms were captured in the context of the study, and therefore obfuscate domain specialisms.

**Table 1. Interview participant summary.**

| Code | Role | Sector | Self-reported Specialism |
|------|------|--------|--------------------------|
| P1 | PhD year 1 | Academic | AI for Physics |
| P2 | PhD year 3 | Academic | NLP |
| P3 | Postdoctoral | Academic | NLP |
| P4 | Postdoctoral | Academic | NLP |
| P5 | PhD year 3 | Academic | NLP |
| P6 | Postdoctoral | Academic | CV |
| P7 | PhD year 2 | Academic | NLP |
| P8 | PhD year 4 | Academic | NLP |
| P9 | Data scientist | Academic-related | NLP |
| P10 | Data scientist | Industry | CV |
| P11 | Postdoctoral | Academic | CV |
| P12 | Data scientist | Industry | NLP |

**3.2.2 Method  Semi-structured interviews.** Semi-structured interviews are an established technique for collecting data [87]. Prior to formal commencement of the study one pilot user was taken through using a preliminary interview script, which led to the refinement of the interview materials. The topic guide, graphic stimuli, and full transcripts are available online [88]. The topics cover:

- Use of diagrams when communicating research
- Use of diagrams when consuming research

Following the graphic elicitation and card sorting exercises detailed below, additional follow-up questions were asked, about the role of diagrams more generally and exploring topics that came up during the interview. The entire interview session, including the two exercises, was audio recorded and documented in the transcript. Six participants were interviewed face-to-face, and six were interviewed over Skype [89] video software. Interview resources were presented as printouts or as PDFs. The interviews took an average of just over 1 hour, resulting in 12 hours, 4 minutes, 54 seconds of audio recording. The recordings were transcribed with personally identifiable information and unnecessary non-words redacted, resulting in over 58,000 words of transcription. The interviews were conducted in English, and the majority of participants were non-native English speakers. The transcripts capture what was said, with the interviewer adding clarifications of understood meaning in square brackets where required. **Graphical Stimuli** Graphic elicitation is a complex term, used in a variety of ways, as discussed by Umoquit et al. [90]. This study utilises pre-made diagrams as stimuli, fitting with Crilly et al.'s [91] definition and usage of graphic elicitation. In this setting, example diagrams are used for "graphic communication" rather than "graphic ideation". Graphic stimuli were chosen to be used as part of the interview for the reasons outlined by Crilly et al.: That it allows a shared frame of reference, facilitates complex lines of enquiry, and provokes comments on interpretation and assumptions.

The six example diagrams used were chosen after the research team conducted an open card sorting exercise on a manually extracted corpus of 120 scholarly NN system diagrams. Twenty diagrams were randomly selected from each of CVPR 2019, ICCV 2017, ECCV 2018, ACL 2019, NAACL 2019 and EMNLP 2018. From these 120 diagrams, six groupings were identified: "Labelled layers", "3D blocks", "Pictoral example centric", "Text example centric", "Modular" and "Block diagram". The groups are not distinct, but encompass the main visual aspects that authors seem to be prioritising. This follows the classification advice of

Futrelle [92], stating that "family resemblances" are often the best that is possible for diagram schemas. The specific examples used were selected from 2019 conferences, and were chosen (a) to be contemporary, (b) to cover a range of venues, (c) to be visually different and (d) to be clearly placed within the groups identified. This selection criteria was chosen in order to cover the search space, and facilitate discussion about the different visual and content aspects. Due to the heterogeneity of representations used in the field, it was not felt possible to construct a small subset that were truly representative of the heterogeneity whole field. Instead, the examples were chosen with the aim of covering a range of the most commonly observed diagrammatic phenomena. Depending on whether the interview was in person or online, the stimuli were presented to interviewees on paper or as pngs.

**Card sorting.** Closed card sorting [93] asks users to put cards into groups. The cards used had tasks a researcher might perform using scholarly diagrams, such as "Identifying corpora and data types", "Identifying specific architectural features" and "Memory aid" (see Table 2 for a complete list). The groups used were: "Important in your use of diagrams", "absolutely not important in your use of diagrams, do not do this at all" and "somewhere between". This method was chosen in order to gather quantitative data about reported usage. Initially participants were encouraged to rank all the cards from most to least important, but this proved to be too time consuming for the first participant, so instead these groupings were encouraged. The tasks list was generated based on the experience of the researchers conducting the study, and participants were given the opportunity to add or remove from this list. Depending on whether the interview was in person or online, the stimuli were presented to interviewees as cutout paper slips or a textual pdf document.

**Analysis method.** Thematic analysis was conducted, following the framework of Braun and Clarke [94]. The interview transcripts were imported and themes labelled using NVivo 12 qualitative data analysis software [95]. A bottom-up analysis was appropriate due to the lack of theoretical framework to inform *a priori* categorisation. With a brief commentary, the steps were:

**Table 2. User tasks reported by each participant when reading diagrams. Y indicates a "top three most important task" and N indicates "do not do at all". Some participants did not select precisely three tasks, as explained in Sect 4.3**

| Task | P1 | P2 | P3 | P4 | P5 | P6 | P7 | P8 | P9 | P13 | P11 | P12 |
|---|---|---|---|---|---|---|---|---|---|---|---|---|
| Identifying corpora and data types | | | | | N | | | | Y | N | | |
| Identifying representational choices (e.g. embeddings, graphs) | | | | | Y | | Y | | | | | |
| Identifying the purpose of the system | | | | N | Y | | | | Y | Y | | |
| Identifying specific architectural features | Y | | | | | | Y | Y | | | Y | |
| Identifying opportunity to alter the architecture | | Y | N | N | N | N | Y | Y | N | | | |
| Identifying what the author thinks is important to communicate | N | | | | | N | | Y | Y | N | | |
| Comparing to other systems | | | | | N | | Y | | | | | N |
| Initial check to see if they use a particular thing I am interested in | | Y | | | N | Y | Y | Y | | | | Y |
| Index to navigate the paper | N | Y | | | N | | | N | | N | | Y |
| Memory aid | N | N | | N | Y | N | N | Y | | | Y | Y |
| Aid for writing a summary of the paper | | | Y | | | | | N | | | | N |
| Understanding how the system works | Y | | Y | Y | | Y | Y | | | Y | Y | Y |
| Identifying input and outputs [Participant] | Y | | | | | | | | | | | |
| Parameters to rebuild [Participant] | | | | Y | | | | | | | | |
| Gauge overall complexity [Participant] | | | | | | | | | | | Y | |

1. "Familiarising with the data": Assisted by researchers conducting the transcription.
2. "Generating initial codes": Where applicable, latent themes were chosen rather than semantic/literal, in order to examine underlying issues. Initial scope was the entire interview content.
3. "Searching for themes": In gathering themes, the investigative scope was very broad, so the thematic analysis was restricted to visual encoding mechanisms. This decision was made from a reflexive standpoint [96], as it enables a framework which will be pragmatic and relatively straightforward for diagram authors to implement.
4. "Reviewing themes": Iterative, between research team, getting external input from a thematic analysis expert.
5. "Defining and naming themes": Including establishing a narrative of the research.
6. "Producing the report": This stage involved tweaking themes and reviewing previous codes, particularly on whether to classify (and in doing so quantify) parts of the qualitative feedback, and identifying aspects to report.

Topics such as diagram content requirements and inter-participant agreement were also captured through the coding iterations. These are presented as categories rather than themes, as the possible responses were relatively restricted ("do you like or not like this diagram", and card sorting of tasks). The thematic analysis supports the research question of which presentation aspects are helpful or confusing.

### 3.2.3 Limitations

- Participants were known to the research team, which could introduce selection bias.
- This is a small scale interview study, and participants were selected to have varied levels of experience and expertise (Table 1). This method is useful for providing rich perspectives from individuals, and allows for a wide range of topics. However, it is a limitation of this approach that findings cannot be generalised.
- Diagrams are considered outside of the paper context, in order to focus the discussion on the diagrams rather than the content of the paper. We mitigated this risk by using example diagrams that were relatively self-contained. This methodological choice, combined with the unnatural interview setting, means we are able to discuss "reported usage" rather than "actual usage".
- Participants took part in the study knowing it was about diagrams, and may be unrepresentatively positive about their use.
- Participant opinion on diagrams may have been distorted by their own perceived self-efficacy: *"So, since I come from a more similar field and I understand this diagram well, I like this diagram."* (P7). We did not assess the correctness of their statements or sentiments as part of this study, in order to help the participant feel at ease. It is possible participants were wrong in their assumptions about the systems, and the accompanying text was not provided to assist in validating assumptions. This could have influenced their opinions on the example diagrams.
- For the card-sorting, we used a broad range of tasks, all of which were relevant and therefore useful to include. We constrained the number of tasks to the level we felt appropriate in order to make the task feasible for participants to undertake. In hindsight, due to the heterogeneous usage of diagrams, there would have been benefit from more precision particularly in "understanding how the system works", for example the level of granularity the user requires.

### 3.3 Framework evaluation

**3.3.1 Participants** Participant recruitment criteria was having created a neural network system diagram "potentially suitable for submission to a computer vision or natural language processing conference". Potential participants were contacted by word of mouth and via gatekeepers. Recruitment took place from 23rd June 2020 to 24th January 2021. Written consent was collected. Ten scholars participated in the study, in two cohorts of five.

This study was conducted virtually, using email, SelectSurvey (data collection) and zendto (secure document distribution). This was not the original design, which was to collect the same data in a 2hr workshop format. This study was redesigned to accommodate restrictions required by the COVID-19 pandemic.

**3.3.2 Method** Fig 4 provides an outline of the method, comprised of multiple stages in order to capture data from multiple perspectives. The design ensures control of participants' access to information, reducing potential bias and confounding effects.

**Stage 1: Framework usability (editing diagrams according to the framework).** After submitting an initial diagram via email attachment, the participants were sent the Framework (Table 6) and asked to edit the diagram according to the framework if they wished. As well as ensuring participants were sufficiently knowledgeable about NNs to meaningfully contribute to the study, the early submission of an edited diagram in Stage 1 served as a pragmatic filter for participants unlikely to proceed through the multiple stages of the experiment.

Following the edited diagram submission via email, participants were allocated into groups of five, based on their submission time: The first five participants to submit a diagram formed the first group. Within each group, participants were allocated a pseudonym, based on their submission time (P1 was the first confirmed participant, and so on). Conducting the experiment in small groups simplified recruitment and allowed faster experiment cycles as the number of people being waited for was limited. This is a useful feature of the method, as it would enable iterative improvement of the framework between cycles. The method was administered entirely online, though the same method could be used in a physical workshop format.

Participants were then sent an online survey via email, containing questions about the experience of using the framework, both quantitatively and qualitatively. Participants also completed a checklist for "which guideline did you use?", with free-text to comment on each. Authors were also asked to describe "how does the system work?". This correctness-defining question was asked after all the diagrams were created, in order that the only intervention was to show the framework. In the experimental design, eliminating the risk of this confounding was felt important, as one might expect that a clearer communicative intent (expressing their communicative goal in natural language as well as diagrammatically) could result in authors making changes unrelated to the framework.

To summarise, in conducting Stage 1, the following data was collected digitally for each of the two groups, each with a participant pseudonym linking the data:

- 10 diagrams of a participant-created system
- 10 diagrams of a participant-created system edited using the framework
- 10 system author summary paragraphs
- Qualitative author feedback on the framework overall
- Feedback and "votes" from author perspective on each guideline item

**Stage 2: Framework utility (interpreting and evaluating diagrams, and qualitative feedback).** Stage 2 focuses on the readership properties of diagrams, in doing so evaluating the impact of the framework on readability. Stage 1 collected 20 output diagrams, comprised of two groups of five systems with two diagram versions. For simplicity of administration, the

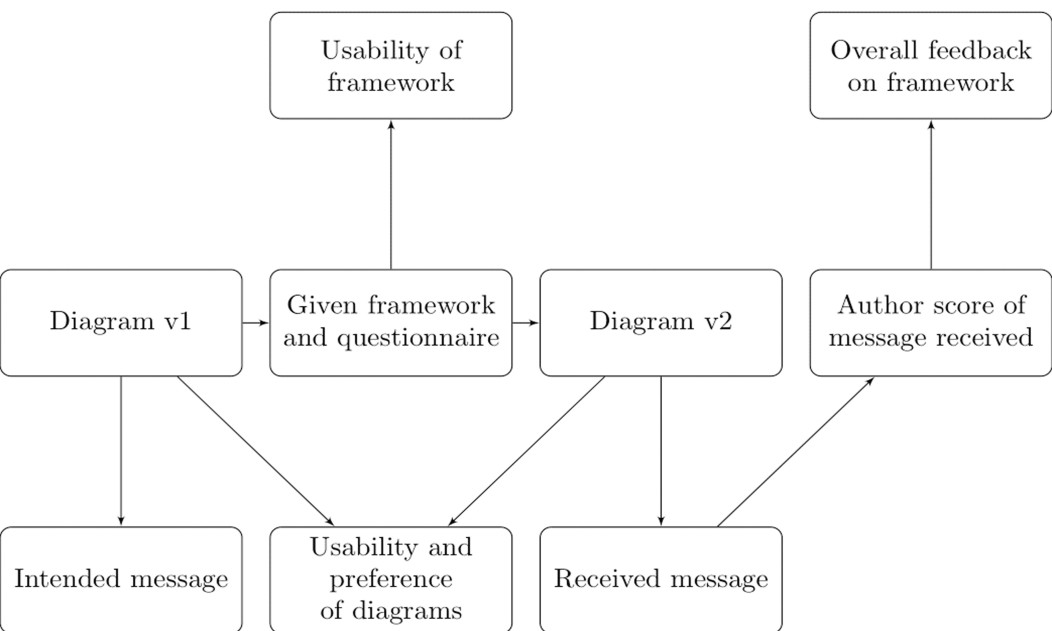

**Fig 4. The two parts of the experiment, completing the communication loop and assessing NN system diagram usability, while facilitating remote administration.** The method is based on accessibility guideline effectiveness investigation [81], with an ecologically valid context and additional quantitative and qualitative measures. The bottom row relates to readership, the middle row to the authorship or diagrams themselves, and the top row to the framework (though some aspects are shared across different streams).

same participants were used in the role of readers. This reflects the use in scholarly practice, where scholars both author and read diagrams.

Within each group, each participant was given the other 4 systems to review in a digital form, distributed via email. The ordering was random, and excluding their own.

- For each system, at random, participants were either shown pre- or post- framework diagram first.
- Participants were asked to write down "how does the system work?".
- Participants were then shown the other version, and asked which of the two they preferred, and given the opportunity to provide textual feedback about the diagram.
- Within each group of 5 participants, each pair of diagrams was reviewed by 4 other participants. With 10 participants in total, this led to 40 reader-created summaries corresponding to 10 author-created summaries.

The reader feedback submitted for each diagram was collated, anonymised, and distributed by email attachment to the original author for their 1-5 assessment on correctness, guided by the author summary paragraph (the author's "intended message") created in Stage 1.

**Limitations.** Usability was measured by author's propensity to use the framework, as well as the formal report by authors on whether they felt it was a useful guideline. In terms of propensity to use, the usability of the framework may appear to be lower than may be expected, as in this study the participants had necessarily already been through the process of initially creating the diagram without the framework. In terms of propensity, the usability of framework is evaluated in the context of editing a diagram. Usability in the context of

creating a new diagram was not evaluated: Whilst this may be a legitimate use case, it does not facilitate A/B testing of the impact of the framework.

Diagrams are examined out of their textual context. This is a limitation, particularly since scholarly publications use multiple media which may play different and complementary roles in learning and information acquisition [97]. However, the interview study found that some readers use system diagrams in NN papers as the primary source of information about a paper. As such, this limitation is noted but improvement of these figures distinct from text is still worthwhile. Related to this, while viewing specific diagrams, 3/10 participants in this study expressed difficulty in considering the diagrams without text (not for the same diagram). All participants were permitted to include a caption for their figure, if they wished (2/10 participants included a caption).

Participants were not informed which of the diagrams they were viewing were pre- or post- framework, which reduces potential bias and simultaneously constrains the ability of the assessor to evaluate the framework.

Whilst being holistic, this study does not explicitly assess propensity for user adoption. The framework does not exist in isolation, and adoption is likely to be challenging, as has been found in publication guidelines for clinical epidemiology [98], and might be reasonably expected to depend on other factors. For example, the creation of a framework-compliant diagramming tool might be expected to increase framework adoption. This study aims to stimulate development and discussion about improved NN diagrams by making them the subject of study. The framework is not intended as a final output, but rather a step on a journey for supporting better diagramming practices. As such, evaluation of user adoption was not prioritised.

In scoring the correctness of diagram understanding, omissions in understanding cannot be captured, and many of the descriptions, both by readers and authors are quite brief. This is mitigated to some degree by allowing free text in the reader diagram assessment to highlight omissions or potential confusion.

This is a small scale, in depth study, and results may not be generalisable to neural network subdomains which are not specialisms of participants.

**Overall measures.** Due to the involved nature of this study, the measures collected are summarised.

Stage 1, with a focus on usability of the framework:

- Author: Comments on the framework overall. This measure is to capture qualitative feedback, and encourage reflection on the set rather than the individual guidelines.
- Author: Comments on individual guidelines. This measure is to gain qualitative insight around individual guidelines, and to surface unintended consequences of wording.
- Author: Votes on individual guidelines. This quantitative measure is designed to capture whether the individual guidelines are useful, and to aid in prioritisation within the framework.
- Author-Pair-of-Diagrams: Diagrams "before" and "after" exposure to the framework is captured. It is also reported whether authors chose to edit their diagrams in any way. This is not used as a measure in this work, but may support further analysis.

Stage 2, with a focus on utility of the framework:

- Reader-Diagram: Author rating on reader summary (/5). This measure is intended to capture communicative efficacy of the diagram, using a textual summary as a proxy for

understanding. This is split into perceived correctness and confidence in the two subsequent measures, in order to capture any uncertainty in the author's rating.

- Reader-Diagram: Reader correctness as assessed by author.
- Reader-Diagram: Author confidence in the correctness of their assessment.
- Reader-Diagram: Perceived quality (/5) of individual diagrams by reader, with comments. This measure captures whether the reader thinks the diagram is effective, and is designed to aid debugging of any communicative gaps.
- Reader-Pair-of-Diagram-Opinions: Reported preference (A/B), with comments. This measure captures the impact of exposure to the framework on reader preference, a partial measure of diagram quality.
- Participant-Overall: Comments by author having seen whole process (including being a reader). This qualitative feedback is captured to allow reflective comments about the framework and the experiment itself.

These measures are designed to be holistic and capture different dimensions of the diagramming process, and capture quantitative data alongside qualitative data to facilitate richer explanations.

Additionally, participant attribute data was collected (such as experience, background sentiment about diagrams, and institution).

For Stage 1, the extent of the visual difference between what is drawn before and after exposure to the framework is not examined. This decision was taken because of the plethora of possible ways to do this, from pixel changes to cognitive properties. Though this visual difference may be relevant, the qualitative feedback was deemed sufficient and unambiguous to collect and summarise.

For Stage 2, note that time on task is usually analysed in diagram usage experiments, but is not done here. As a measure, it is unreliable and more importantly is not the thing being optimised for here: For NN system diagrams, efficiency is much less important than effectiveness. In general to use this method, the metrics should match the desired outcome of the guidelines. For example, for accessibility guidelines efficiency could be more important.

The primary goal of this study is to evaluate the framework itself. Whilst data has been collected related to the diagrams themselves, and the participant answers, analysis is focused on the impact of the framework. Analysis targets three hypotheses:

- Hypothesis: Post-Framework Diagrams are on average preferred to Pre-Framework Diagrams.
- Hypothesis: The majority of the guidelines will be reported as useful by the majority of authors.
- Hypothesis: The framework will be recommended by most participants.

### 3.4 Corpus analysis

This method is outlined in Sect 6.2, since it utilises the results of the previous studies.

## 4 Results I: Interview study

This analysis examines the differences in opinions and usage of NN diagrams. Transcripts were uploaded into NVivo 12 qualitative data analysis software [95]. Reporting of this study is centred around the research questions, and includes thematic analysis based around user requirements for reading diagrams. The reporting does not reference individual diagrams in each instance, because (i) many quotes are not in reference to a diagram (ii) some diagrams

have multiple instances of a phenomenon, which the participant may be referring to only one of, and (iii) the transcripts are available and are easily text search-able for the quote, with the example diagram letter added where it is not easily identifiable from the verbal transcription alone. Direct quotes from participants are reported in italics to improve the readability of this chapter.

## 4.1 Why do NN researchers create diagrams when authoring papers?

Participants reported constructing diagrams in order to give a *"schematic overview"* (P1), to provide an *"anchor"* (P12), and to *"simplify understanding"* (P3). Whilst these topics are conceptually intertwined, analysis led to three categories of why people use diagrams: (i) Summary view (ii) Perceived effectiveness, and (iii) Relation expression.

**Diagrams as a summary (IW1).** All participants expressed that their diagrams (or those they read) facilitate a holistic overview of the system. This also relates to the use of the diagram to screen for a paper's relevance to the reader.

- *"Because in the diagram you can express directionality and you can show consistent things over the whole approach. You can have a holistic view in a diagram, that is quite hard to do in text, there is a bunch of stuff you have to go through to understand. For a new reader, providing a diagram that gives you most of the picture."* (P7)
- *"I use a diagram when I want to summarise or represent a higher level view of some process, or the building blocks of a method that I'm trying to convey."* (P8)
- *"So what I prefer to do through the diagram is give the intuition rather than the specification."* (P12)

The difference in opinion on whether a diagram should provide a schematic summary appears to be linked to the macroscopic readership behaviours described in Sect 4.1, and is further complicated by the level of "engineering" specification the participant felt beneficial to be detailed diagrammatically.

Participants reported part of the value of the diagram being in the omission of information in order to make understanding easier for readers, overlapping "summary" with "relational communication": *"there are some things that a diagram can just explain really succinctly and clearly"* (P11).

**Diagrams are perceived as effective compared with text (for relational communication) (IW2).** All participants expressed that diagrams were good, useful or effective for communicating systems, though this was often latent or with vagueness around the reasons for this (e.g. *"I like diagrams"* (P1), *"Diagrams are good"* (P2)).

The perceived effectiveness of diagrams led two participants to compare the utility of text and diagrams. P3 described text as being an effective default modality, with diagrams secondary: *"Maybe this is the type of information that reading from a diagram is more effort than reading from a text. The main idea of creating a diagram is to simplify the understanding. If the concept is easy to describe in a few words, it is better to read a paragraph than to look at a picture and try to decode it."* (P3). In contrast, P4 described a diagram as being an effective default modality, with text secondary: *"Usually I prefer to use a graphical representation because it reduces the time to understand the idea that is being expressed. Text for me is only to explain something that is not easy to do with a diagram by itself."* (P4). Both participants had the suitability of representation underlying their comments, which otherwise seem to be based on personal preference. Overall, the perceived relative effectiveness of text or

diagrams seems to be contextual and personal, as found in many other aspects of the interview
analysis.

The ability of diagrams to represent relations was expressed: *"people can easily lose track of what connects to what and why, while a diagram gives something like a grounding or an anchor"* (P12). Sect 4.4 includes commentary on examples of usage of the relational nature of diagrams, especially navigation, which can be considered as the ability to easily transition between objects using relations. Sect 4.5 explore effectiveness of diagrams in more detail, particularly within the theme "Visual ease of use".

**Diagrams are relational (IW3).** Participants saw the (relational) complexity of their systems as better represented by a diagram than linear text. The comments on this topic focus on explanatory and communicative value. Participant quotes express this: *"It is not easy to explain a complex network without a diagram."* (P4) and *"I am not sure they would grasp this concept of compositionality as well through text…what I try to do is show how this fits together."* (P12).

For some participants there were multiple reasons for using the relational advantage of diagrams: *"It encodes a couple of things quite well, it encodes data flow and also computation steps, so it is a nice way of doing both of those two things at the same time. [Pause] It is generally easier to walk through people, so if you are presenting some design it tends to be easier to walk people through what is going on by pointing at blocks and describing that particular block in the overall picture."* (P10).

**Diagram relation to the text (IW4).** This focuses on *readership* usage, though readership may be considered as intertwined with authorship. In an academic paper reading context, six participants reported using diagrams as an accompaniment to the text, while six reported a special role for diagrams.

3/12 participants reported reading the diagram before the text of a scholarly publication. *"I personally start with the diagram to get a general view"* (P3), *"Even before reading the abstract or conclusions. I go directly to the diagram, this is what I'm looking for"* (P6) and *"Reading the paper starts with the diagram, for me."* (P7). This contrasts with the comment that *"It is a process of moving to and from the text and the diagram to get a complete picture."* (P1). It is interesting to note, particularly in terms of usability, that these three participants expressed time pressure as the reason for using the diagram in this way, and were using the diagram as a cross-cutting schematic to understand and screen the paper for relevance. This usage of system diagrams suggests that the diagrams may be fulfilling a role as a type of informal graphical abstract.

An initial overview was not the only special role afforded to diagrams. Participants also reported the diagram as a primary view into the scholarly paper *"if I look at a diagram and I really can't figure out what this is doing, I'll usually read the paragraph before, or I'll scan the text around it to see where the figure is referenced and read that bit. I don't have enough time to read whole papers."* (P10) and as an index *"I go back every now and then to the diagram to see how it fits together."* (P12).

## 4.2 How do NN researchers create diagrams?

**Intuitively, inspired by others, or using their own set method.** Six participants reported not having a systematic method to create diagrams. Two quotes describe this unsystematic authorship process: *"There is no standard for diagrams in our area, so I try to make something that makes sense, or seems to make sense, and additionally put some explanation and hope the reader will understand."* (P3) and *"I guess I've always got an idea about what a figure is trying*

*to communicate that I feel is easier done in images than words but it just depends what that happens to be.”* (P2).

Three further participants’ methods were to use *“the people we cited; their diagrams”* (P5) to design their own diagrams.

Two participants had created their own standard method for authoring diagrams, and both were inspired by related work in the creation of this method. One (P8) had a method of always using labelled layers, the other (P10) created a custom diagram notation: *“I do enforce that when I’m doing my own notebook diagramming, I’ll use a consistent set of conventions for myself…the abstraction just melts away, and you can understand what’s going on rather than the shapes.”* (P10).

One participant (P9) was not authoring papers for conferences.

**User tasks when creating diagrams.** The interview focused on authoring and readership practices for scholarly publication. Some additional uses of diagrams arose during interview.

In addition to being used in the task of authoring of a paper, all non-academic researchers described the use case of giving presentations that was not commonly discussed by the academic participants.

When asked about creative process, participants mentioned using block diagrams in a broader cognitive context, to understand their own wider projects (P3), to solve coding problems (P1), and to interpret papers (P4). For some participants diagrams were fundamental in their research process: *“Interviewer: Do you draw diagrams then before you are building the system? P4: Before, during and after [Laugh]. The whole process.”* Five participants mentioned having an iterative diagram creation process, often including a colleague or supervisor, or using a whiteboard.

**Software used to create diagrams.** The software tools used to create diagrams were also very diverse. Between them, the 12 participants reported using 16 different digital tools to create NN system diagrams. Fig 5 shows the distribution of count of tools used. The main reasons for tool choice were ease of use and file export format. Inkscape (4), Google Draw (3), Draw.io (2), and TikZ (2) were the most commonly used tools, with other tools used by only one participant (astah, fast.ai, Google Slides, graphviz, Illustrator, Lucid Chart, Omnigraffle, Open Office Draw, Microsoft Paint, Microsoft Powerpoint, Microsoft Visio, and yEd). Six participants commented that a custom tool for creating NN system diagrams would be useful, either as a plug-in to an existing tool or stand-alone.

## 4.3 What tasks do system diagrams support for readers of scholarly papers?

The card-sorting exercise gave insight into the tasks performed by participants reading NN systems architecture diagrams. It had been expected that some tasks would emerge as prevalent. However, we discovered heterogeneity. In this exercise, we started with 15 core tasks, as shown in Table 2. During the activity, three participants mentioned further tasks they perform which were not included in the presented list. These have been added to Table 2 and labelled “[Participant]”.

Participants were asked to pick their top three tasks, and to highlight any number that they felt they did not do at all. The results are reported in Table 2. Five participants chose to group some tasks together into one task, and one participant only chose two top tasks, leading to non-conformity with “three top tasks” for those participants. The researchers were intentionally not rigid in enforcing the limit, as the intention was to understand usage rather than force participants into the task framework. P11 did not select an “do not do at all” task, stating *“You*

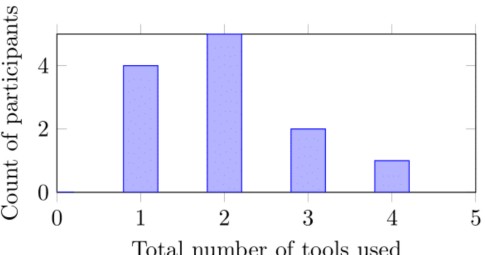

**Fig 5. Number of different digital diagram creation tools reported as being used by each participant.**

*wouldn't do all of them because then you'd be trying to put too much information in the diagram. Depending on what you're trying to get across, each of these could potentially be uses. I wouldn't rule any of them out".* All 12 presented tasks were chosen by at least one participant as a "top importance" task. Nine out of 12 tasks were reported as "not done at all" by at least one participant. This highlights the variable use of diagrams by readers. The top-rated tasks were:

- "Understanding how the system works" (8/12)
- "Identifying the system novelty or contribution of the paper" (5/12)
- "Identifying layers, relations between components, or internal dependencies" (5/12)

That these modal tasks were selected by so few participants further underlines the differences in users' requirements of the diagrams, and complements the differences in opinion seen when discussing the Examples. To quantify this, Fleiss' Kappa for m Raters was executed in R software (Subjects = 15, Raters = 12). This gave $\kappa = 0.121$ and $z = 3.79$ with p-value = 0.000149. This indicates, with significance, that there is only "slight agreement" on top tasks. The low number of added tasks, and that each core task was chosen at least once, suggest the 15 core tasks presented have reasonably good coverage of the task-space.

## 4.4 What aspects of presentation do researchers find helpful or confusing?

Generating codes was done using the thematic analysis protocol described. However, due to the need to articulate the heterogeneity uncovered, the clustering aspect of thematic analysis was not appropriate in the reporting of this section. To futher clarify, this subsection contains codes that are categorical labels, not themes. When describing requirements for the domain in Sect 4.5, full thematic analysis is used. First, content-related aspects are reported, followed by presentation-related aspects.

**Schematic overview versus implementation details (IP1).** In terms of what the diagram should convey, nine participants expressed that the system diagram should provide an overview. One participant commented that diagrams should be context dependent (P11), and two participants required implementation details such as hyperparameters (P1, P4) which 10/12 participants deemed unnecessary. When prompted with example diagrams, participants occasionally changed their preferences. This highlights the benefit of graphic elicitation.

**Use of concrete examples (IP2).** Two participants used diagrams as a way to instantiate an example (P5, P12). Ten participants reported finding an example input helpful (not P4, P7). The instantiation described was key for the two participants' cognition (P5, P12): *"I tend to inductively understand something. That is, from an example, generalise to how it works...They usually don't put examples in text, the example is usually in the diagram"* (P5).

**Technical knowledge and familiarity (IP3).** All participants commented that technical knowledge was required to understand the example diagrams. Comments ranged from *"Maybe this is easy to understand because I know what resnet is."* (P4, overall positive opinion) to *"I've not used resnets so I don't know what resnet-34 means"* (P11, overall negative opinion).

Familiarity appears to have had a substantial impact on opinion of the diagram, for all participants: *"Since I come from a more similar field and I understand this diagram well, I like this diagram."* (P7). Preference difference is quantified in Sect 4.7, though (with the small sample size and broad specialism groupings) no clusters were identified, despite investigation into any similar sentiment based on primary research domain.

The following topics relate primarily to visual presentation choices, rather than content.

**Navigation (IP4).** Diagram E, which features labelled modules containing further detail, had conflicting views on the ease of navigation: *"Yeah this is automatically easier, because you've got the input on the top left, and it's a bit clearer about where you're supposed to follow the model, as you've these clear arrows and guidelines."* (P1) contrasts with *"This lack of direction here. I like how they have put examples here, so at least I have a start and an end, but there is a lack of directionality in the diagram."* (P7). These two quotes reflect the differences in perception and perceived meaning found elsewhere in this study.

**Ineffective use of diagrams relational properties (IP5).** Commenting while reading example diagrams, 2/12 participants suggested text would be more appropriate than a diagram *"It's nice and linear, but has not revealed much more to me than the text caption did to be honest."* (P10) and *"The diagram isn't giving you over and above what you could get from text. So in the sense of conveying information clearly, because it is that simple and basic it isn't giving you anything a simple sentence wouldn't give you."* (P11).

**Precision meaningfulness (IP6).** Participants expressed varying opinions as to whether a set of circles related to a specific or an arbitrary dimension of tensor. This style of depiction of a tensor is fairly common in neural networks. Table 3 demonstrates this issue through some of the conflicting quotes. The comments are based on Example A or E.

**Hyperparameters (IP7).** A final comment on specificity relates to hyperparameters, the settings for the system that are used to control the training process. This includes, for example, the size of the tensor used in each hidden layer, and is therefore related to the requirement for precision, but not to the visual encoding of precision: The dimension of a hidden layer is often quite large. In practice, few authors attempt to visually encode hyperparameters of their systems, relying instead on labelling using numbers (as in Example B). Feedback on explicit numerical representation varied from *"If you want to see that in an experimental section, they are not fixed. I would not expect to see that in an architecture diagram, certainly not."* (P8) to *"It would be good to have a labelling of the dimensionality of the different layers"* (P1). The example of hyperparameters is indicative of the varied levels of granularity required by participants. This is not always the case: At a similar level of granularity, but often required by participants, was the detailing of specific functions such as "loss" or "pooling".

## 4.5 Visual encoding user requirements: Thematic analysis

**4.5.1 Introduction** The thematic analysis focuses on visual encoding user requirements, and as such examines preferences, which are primarily explicit rather than latent. Despite the variety and conflict of opinions demonstrated previously, there is some underlying commonality. This subsection simultaneously considers both the creation and consumption of diagrams. The top level themes identified were visual ease of use, appropriate content, and expectation matching, with multiple sub-themes. Codes in this subsection are in addition to the codes and categories outlined in the previous sections.

**Table 3. Participants expressed a spectrum of opinions, of differing confidence, on whether a precise depiction is meaningful. P7 made two different comments, and P3 and P9 did not make comments mapped to this theme**

| Not meaningful | Need to refer to text or unsure | Meaningful |
| --- | --- | --- |
| P1: "I doubt it, I wouldn't expect so." | P2: "Well, it's not clear whether that is just an abstract representation, I'd have to look in the text." | P6: "Yes! I mean this is computer science not literature. If you have four, it means you have four." |
| P7: "And the number, it cannot be four dimensions, it must be much more than that. It is either misleading or just plain their own internal reasoning to put it like that." | P4: "I don't know." | P10: "So we've got the embedding layer which is a 5-vector and out inputs which are 4-vectors." |
| P8: "No. I don't think they are significant." | P5: "It might be significant or it might be arbitrary, we'd probably need to check in the paper." | |
| P11: "I'm working on the assumption that there aren't four inputs there, and it's kind of an arbitrary number." | P7: "Or maybe the hidden layer has some other transformation layer, then maybe that is why they put that. I'd go in and check it out in the paper." | |
| P12: "And the fact that they have four and five, I think probably the reason why they did that was to show that the dimensions don't have to match. But I don't think it is that important, to be honest." | | |

**4.5.2 Theme: Visual ease of use** All participants mentioned aspects related to ease of use. Navigation and Ineffective use of relational properties support this theme (Sect 4.4). **Clear navigation (IV1).** Navigational issues were focused on overall structure: *"it is relatively easy to follow due to the structure of it."* (P2). See Sect 4.4 for more detail. **Aesthetics (IV2).** 8/12 said aesthetics are important, in terms of clarity of the diagram. As such this makes the concepts of aesthetics and content intertwined. Aesthetic-related comments covered topics such as graphical components, colour, orientation, "prettiness" and inconsistency.

**Consistency within diagram (IV3).** Consistency was seen as important by all participants. This was sometimes at a structural level (e.g. *"It should be all at the same level of abstraction"* (P6)) and sometimes at a visual component level (e.g. colour inconsistency *"annoys me a little bit because it is making me think that those things are different while probably they are not."* (P12)). The importance of consistency can also be inferred by participants requesting diagram guidelines, frameworks or standards (see Sect 4.6). There is a slight dissonance between the reported importance of consistency and the lack of confusion due to precision meaninglessness (Sect 3). This may be partially explained by the lack of ability to validate assumptions against the full paper. Two participants expressed frustration at having to turn their heads to read rotated labels, saying *"I don't particularly like having to turn my head to read the labels."* (P10), and *"I don't like that some things are written horizontally, some things vertically, because I have to turn my head around to actually read them"* (P12).

**Process stages (IV4).** Particularly when creating diagrams, 8/12 participants commented on using the diagram to understand the process steps *"I need to write something in the paper to not only imagine but to think well about the paper, about sequence, about its choice"* (P4).

**4.5.3 Theme: Appropriate content** This theme builds on the codes of Sect 4.4. **Wanting more information in the diagram.** The "missing" information participants sought includes symbols (P1), what to focus on (P2), specific details *"score is not clear, it would be good to have*

*an explanation"* (P3), maths *"$x_1, x_k, x_n$, this part is confusing"* (P4), inputs and outputs (P8), caption, key or legend (P9) or the purpose of colour (P10).

**Wanting less information in the diagram.** This does not conflict directly with the previous sub-theme, being about presenting the right information. *"I don't know what's important here because everything is on there"* (P2), and *"It is not explicit what is core in the diagram"* (P4), and *"The whole point is it is meant to be concise and get the information over to you quickly but that it not concise at all."* (P11).

**Wanting multiple diagrams.** Eight participants expressed wanting multiple diagrams in order to get different content from each, usually one schematic and one or more detailed for specific components. *"I think we need another, more detailed graph to represent the architecture of the model. This is just an overview."* (P6).

**4.5.4 Theme: Expectation matching** This theme is primarily latent. It encompasses sub-themes relating to social and contextual aspects (Sect 4.1), including familiarity (Sect 4.4), consistency, and more broadly "seeing what I expect to".

**Consistency across diagrams (IE1).** This includes comments on complying with conventions. Half the participants said it was difficult to understand a given author: *"We have to unify the language of diagrams because I take a long time understanding the visual language of each author."* (P4). Ten participants commented on the current lack of a standard visual language. This is further supported by the request for guidelines (see Sect 4.6).

**Consistency within domain (IE2).** This includes numerically representing hyperparameters in CNN diagrams, which is a common practice. Another example of consistency commented on was the usage of domain-specific terminology, such as "resnet-34", "conv1", or "BERT", which are conventional abbreviations for architectural features and seen as useful labels.

**Unexplained symbols (IE3).** These often caused frustration (9 participants). *"There are some links which aren't explained. There is quite a lot of notation, mathematical notation, which is in the diagram but not explained in the caption. I don't know what half, any, of these symbols stand for."* (P1). For other participants this was less of an issue or was case-dependent *"I don't know the symbols like $H_g$. But I assume they are described in the text."* (P6).

## 4.6 Requests for guidelines

Guidelines for creating diagrams were requested explicitly by five participants: *"A set of guidance, something like that, could be super useful for researchers because most people don't really know what they are doing and don't know even basic things about use of shape and colour and fonts."* (P11). The nature of this request ranged from design topics to standardised symbols. All participants made a comment that could be viewed as supporting the creation of guidelines for authors. One participant (P9) requested guidelines for reading diagrams.

## 4.7 Overall diagram impression

This subsection refers to the subjective preference of each example diagram. As such, it may involve any contextual or non-contextual factors personal to the reader. This is not an attempt to discover "good" diagrams, particularly as the diagrams are (by definition of being in a scholarly publication) describing different systems and different contributions. Again, the analysis suggests heterogeneity.

As part of the interview, participants were asked a binary "do you like or not like this diagram" for each of the examples (see [88]). Neutral sentiment was permitted, giving three possible ratings. Reliability of agreement for diagram preference was analysed using Fleiss' Kappa for m Raters, the standard measure of agreement for categorical ratings. This was executed

**Table 4. Overall opinion of example diagram: + = like, - = dislike, blank = neutral.**

| Participant opinion of example | A | B | C | D | E | F |
|---|---|---|---|---|---|---|
| P1 | + | | - | - | | - |
| P2 | + | - | - | - | + | + |
| P3 | + | | + | - | | + |
| P4 | + | + | - | + | + | + |
| P5 | - | - | - | + | | + |
| P6 | - | + | - | - | + | + |
| P7 | + | + | | + | - | |
| P8 | + | + | - | + | + | + |
| P9 | + | - | - | - | - | + |
| P10 | + | + | + | + | + | + |
| P11 | + | + | - | - | - | - |
| P12 | + | - | - | + | - | + |

using R software (Subjects = 6, Raters = 12) giving $\kappa$ = 0.094, $z$ = 2.33, with p-value = 0.02. This represents "slight agreement" between participants, with significance [99]. Table 4 shows all reported overall opinions. Overall, example diagrams A and F were most "liked" (10 and 9 participants liked, respectively), Example Diagram D polarised (6 liked, 6 disliked, 0 neutral) and example diagram C was least liked overall (2 liked, 1 neutral, 9 disliked).

Efforts were made to obtain a higher rater agreement. "Expressed positive" or "Expressed negative" were also examined individually, effectively removing neutral responses. This gives approx 0.1 kappa and p value < 0.05 for either approach, again indicating "slight agreement", with significance. No clusters or patterns of preferences were identified based on role or domain, while noting the small number of participants in this study. This reflects the heterogeneity described throughout the reporting of this study.

## 4.8 Results summary

Diagrams have been shown to be used by authors and readers in a wide range of ways, with a range of needs, perceptions, and preferences. In the interview study, it was found:

- Participants reported a wide variety of tasks performed while reading system diagrams, and had a wide variety of preferences.
- Three important themes were identified: Visual ease of use, appropriate content and expectation matching.
- The usage of diagrams within papers for some researchers is not just as an accompaniment to text, but may be used before (and preferentially to) textual content. This suggests there is some usage of system diagrams as a schematic, or an informal graphical abstract.
- A large variety of different tools are used to author diagrams, even by the same individual researcher.
- Diagram creation guidelines were requested by participants.
- There is potential for confusion and communication error in diagrams being caused by the content and representation. This includes topics such as navigation ambiguity, and whether precise depiction contained meaning.

Table 5 summarises the key findings of this study, including the areas in which heterogeneity was manifest. A possible explanation is that the cognitive tools to support understanding in this domain are still not well developed, so each individual has created their own way

**Table 5. Summary of interview findings related to the usage of scholarly neural network system diagrams.**

| Topic | Observation | |
|---|---|---|
| Heterogeneity | Reading | Perception (e.g. navigation) (Sects 4.4, 4.5) |
| | | Precision meaning (Sect 4.4) |
| | | Use cases (Sects 4.3, 4.2) |
| | | Preferences (Sects 4.4, 4.5, 4.7) |
| | Creation | Method (Sect 4.2) |
| | | Software used (Sect 4.2) |
| Role of scholarly diagrams | Diagrams as a summary (Sect 4.1) | |
| | Diagrams as cognitive entry point (Sect 4.1) | |
| | Diagrams as "example" extraction source (Sect 4.4) | |

of reasoning about the topic, which is manifesting in the diagrams. The fast pace of the field (outlined in Sect 1.1) may also be contributing to this.

This heterogeneity impacts the creation, use and effectiveness of diagrams. The reasons that people use diagrams, how they create them, and their presentation preferences are mixed.

## 4.9 Limitations

- This is a small scale interview study, and participants were selected to have varied levels of experience and expertise (Table 1). This method is useful for providing rich perspectives from individuals, and allows for a wide range of topics. However, it is a limitation of this approach that findings cannot be generalised.
- Diagrams were considered outside of the paper context, in order to focus the discussion on the diagrams rather than the content of the paper. This risk was limited by using example diagrams that were relatively self-contained (understandable without reference to text or other resources). This methodological choice, combined with the unnatural interview setting, necessitates discussion of "reported usage" rather than "actual usage".
- Participants took part in the study knowing it was about diagrams, and may be unrepresentatively positive about their use.
- Participant opinion on diagrams may have been distorted by their own perceived self-efficacy: *"So, since I come from a more similar field and I understand this diagram well, I like this diagram."* (P7). The correctness of their statements were not assessed as part of this study, in order to help the participant feel at ease. It is possible participants were wrong in their assumptions about the systems, and the accompanying text was not provided to assist in validating assumptions. This could have influenced their opinions on the example diagrams.
- The card-sorting used a broad range of tasks, all of which were relevant and therefore useful to include. The number of tasks was constrained to the level felt appropriate in order to make the task feasible for participants to undertake (and was confirmed by the supervisory team). In hindsight, due to the heterogeneous usage of diagrams, there would have been benefit from more precision particularly in "understanding how the system works", for example the level of granularity the user requires.

## 4.10 Summary

Diagrams are an important and widely used way of communicating the architecture of neural network systems. This interview study finds heterogeneity in the way they are constructed and understood, which provides freedom for the author, but leads to potential inaccuracies in their interpretation. Existing HCI guidelines have relevance for scholarly neural network

system diagrams, but no set maps directly to the issues uncovered in the study. To bridge this gap, a framework of guidelines are proposed, specifically addressing the main causes of confusion. Reporting of this interview study is concluded with a participant comment that concisely summarises the findings of this study: *"I think this lack of language for diagrams is so bad, even at a high level there is nothing the same at all."* (P10).

## 4.11 Synthesis of the interviewstudy to create a framework

**4.11.1 Possible approach alternatives**   To explore possible options, the pro's and con's of some of the possible approaches are described below, in order of increasing level of intervention.

1. No intervention. **Pro:** Minimises work done. It is possible, as in other disciplines, that conventions will appear in due course as people copy one-another or sub-optimal conventions are prescribed. It is also possible there would not be normative convergence. **Con:** In the interim, inefficient communication and all authors tackling the same problem independently is also inefficient.
2. Take existing general "design" guidance and share it with NN researchers. **Pro:** Low research effort. Benefit clarity if adopted. **Con:** Not customised for NN domain, not addressing representational priorities or challenges. Difficult to know which framework would be most appropriate for the domain. No tooling to support it. Would require finding ways of engaging NN researchers in visual design.
3. Create a new guidelines-based framework for NN diagrams. **Pro:** Allows targeting to the domain. Easier adoption. Provides a publication avenue to engage NN domain. Potentially enforceable by publishers or reviewers in the same way as other figures are standardised. Allows freedom to express new concepts. Can easily be changed as user needs or representational requirements change, or as evidence emerges about utility of specific guidelines. Interview participants requested this. **Con:** Requires research effort. Would benefit from empirical evaluation or evidence of efficacy. Does not provide all the benefits of a standard language.
4. Create standard NN diagramming language. **Pro:** Large cognitive advantage with widespread adoption. Could be enforceable by publishers, and have supporting software. Diagramming software may have an impact advantage over theoretical work, as they are implementable and more likely to be more visible to those creating the diagrams. Further, software may also have uses beyond the preparation of papers, such as in other diagram presentations or used in further software development. **Con:** Prescriptive and may not be suitable for new approaches. Requires constant maintenance to remain relevant. Would benefit from custom diagramming tools. Would benefit from structured understanding of NN systems. Would have many stakeholders in researchers, practitioners, and ML software tool creators. Requires substantial community building effort, and community engagement (in a representative manner) for widespread adoption and benefits. **Comments:** Note that this approach of scholar-focused diagramming tools integrated with ML software has been taken for CNNs as part of Net2Vis [100]. NN-SVG [101] is designed to support diagramming of a limited set of architectures (FCNN, CNN, and one style of Deep Neural Network), as is the LaTeX code generated by PlotNeuralNet [102]. A list of diagramming tools, representing specific architectures and often subsets of an entire system, is available online [103]. As discussed, the scholarly research area of diagrams has limited visibility in the AI community.

Specifically when comparing with the research supporting Net2Vis [25], despite visibility challenges, the framework:

1. Applies to all NN software, not just those created with Keras or other specific technology stack.
2. Is flexible depending on the contribution's representational needs.
3. Can be applied to a broader range of architectures, such as those with non-CNN components or architectures which do not lend themselves to a linear visual representation.
4. Can be used as an evidence base for the requirements of future diagramming tools.
5. Demonstrates an improvement in preference. Net2Vis does not make a significant difference to answer accuracy in their evaluation.

**4.11.2 Accessibility**   As in many areas, there is a risk of cultural bias in diagrams, which can be reduced in order to increase accessibility, participation and engagement. In an intercultural study of students' diagrams, Deregowski and Dziurawiec [104] attributed improper integration of the objects in the figures for the errors, attending elements in isolation. Given the diversity of authors in ML, including the large and increasing scholarly contribution and practical implementation undertaken by authors and institutions of China [105], it is important that different cultures be consulted and engaged in the creation of any diagrammatic standards.

Through the framework, accessibility is aimed to be improved by:

- Adopting an approach of intercultural collaboration from the outset.
- Creating a diverse community to drive diagrammatic standards forward.
- Advocating individual guidelines which are aligned with existing accessibility checklists, including reducing reliance on colour [106].
- Reducing reliance on English language proficiency to communicate about ML systems.

**4.11.3 Theoretical diagram guidelines**   Having selected an approach, it is helpful to consider existing guidelines for diagrams which could be leveraged in this domain.

In abstract diagram design, there are fragments of advice to be found in the literature, on topics from cognition to perception. Perhaps the most concrete diagram guidelines are those derived by Larkin [107], for maximising the interpretability of diagrams:

- Group together spatially information that is used together, in order to avoid searching during inference,
- Avoid symbolic labels, and
- Make use of perceptual enhancement, for example working from left to right.

Whilst perhaps useful for neural network diagrams, neither grouping nor symbolic labels featured prominently in the interviews, suggesting these may not be the priority for useful guidelines in this domain. Concrete examples of perceptual enhancements applicable to diagrammatic representations can be found in Gestalt laws [108]. Relevant Gestalt principles for AI diagrams include proximity, similarity, closure, direction, and habit (common association). They optimise for perceptual ease, such as easy discriminability of elements. Gestalt laws seem to be useful for consideration, however they are essentially for perceptual effectiveness, rather than communication, and are optimised for visual speed rather than communicative efficacy. Gestalt principles feature heavily in UX guidelines (see Sect 4.11.6).

There are further practical recommendations that can be found in other studies. These offer at least a partial view on "good diagrams", and include ensuring good labeling and highlighting relative importance [109], using non-linguistic symbols depending on audience experience [110] and minimising the number of symbolic elements [111].

These general diagramming guidelines appear relevant to neural network diagrams. However, they have not been designed for the complexity of neural network systems, nor communicative scholarly tasks, and would benefit from empirical evaluation.

**4.11.4 Design guidelines** There are existing design guidelines which may support the improvement of diagrams, such those relating to User Interface Design [112]. Four of Schneiderman's "Eight Golden Rules of Interface Design" are applicable to diagrams:

- "Strive for consistency": Whilst Schneiderman focuses on internal consistency, it was contextual consistency that came out strongly in the interviews. As such, is it unclear how well this translates from interfaces to diagrams.
- "Design dialog to yield closure": Extending this concept potentially leads to inputs and outputs being good to include, as they give more completeness. It also could support the "expectation matching" theme.
- "Reduce short-term memory load": Supports the comments about schematics and simplicity.
- "Enable frequent users to use shortcuts": This supports abbreviations and exploitation of existing conventions.

The remaining four rules are not relevant to (static) diagrams, as they focus on interaction: "Offer informative feedback", "Offer simple error handling", "Permit easy reversal of actions", "Support internal locus of control".

**4.11.5 Information visualisation guidelines** Of Schneiderman's [113] seven tasks "overview, zoom, filter, details-on-demand, relate, history and extract", two (overview and relate) were found to be important tasks for participants. Schneiderman's Mantra of "overview first, zoom and filter, then details on demand" does not appear to be useful for static system diagrams. This perhaps reflects that, whilst the high level may not fit diagrams, there is insight which can be gained from the granular empirical evidence from which these are derived.

Tufte's [45] influential work centred on data visualisation also includes application to information visualisation more broadly, and spans a wide variety of two dimensional representations. In his wide-ranging coverage of domains from multivariate data to planetary relationships, and from maps to music, Tufte comments that for technical engineering diagrams "What matters - inevitably, unrelentingly - is the proper *relationship* among information layers." (emphasis in original). This was also suggested by the interviews.

Several of Tufte's guidelines support creation of schematic diagrams, such as "Maximise Data Ink; Minimise non-data ink" and avoidance of "Chartjunk". Further, support for his guidance on effective use of colour, and emphasising a horizontal direction, can be distilled from the interviews as being sometimes problematic for readers of NN system diagrams. A number of Tufte's recommendations may be less appropriate for complex systems diagrams, such as (a) high density being desirable, (b) assuming everyone is an expert, and (c) giving readers all the data so they can exercise their processing power. This advice would appear to conflict with the aims of the "summary overview" use case indicated by participants (see Sect 4.1).

**4.11.6 User experience guidelines**  User experience (UX) draws heavily on both design and information visualisation, combining evidence and drawing new conclusions with relevance to the UX domain. Hartson and Pyla [114] recommend Tufte's approach: "Don't let affordances for new users be performance barriers to experienced users", whilst suggesting to "Accommodate different levels of expertise/experience with preferences". Hartson and Pyla's guidelines for UX cover a wide field. Many of these guidelines are related to the fields of Design and Information Visualisation, and appear relevant to system diagrams (as discussed in Sects 4.11.4 and 4.11.5).

An important UX consideration is accessibility, the ease of use for specific user groups such as those with disabilities. Accessibility in diagrams appears to not be prioritised, particularly with respect to colour-blindness, language fluency, the examples chosen, or textural or mathematical modalities.

It is not clear without further research whether Hartson and Pyla's guideline to "Support human memory limits with recognition over recall" would also be advisable for scholarly system diagrams. In scholarly research, recall (the retrieval of related details) is required to place this diagram against related work, whilst recognition (the ability to identify familiar information) is likely to aid efficient perception of the diagram. Scientific practices make the use cases for scholarly diagrams more integrated with their context than for a comparatively stand-alone user interface, and as such the prioritisation of recognition over recall may also be different.

**4.11.7 Framework creation**  Few of the existing guidelines are evidence based, and none have been empirically evaluated in a scholarly domain. For the specific cognitive tasks performed in research, and for the specific representational requirements of neural network systems, it may be expected that guidelines devised for general usage would not be appropriate. The framework is created based use cases, visual encoding themes and common confusions highlighted in the interview study.

Table 6 proposes a set of guidelines designed for pragmatic improvement to neural network system diagrams, based on the findings of this study. The intention is for this framework to be adapted as the community's requirements change, and evidence for the efficacy of each guideline is established. Table 7 summarises the links between individual guidelines and the interview study from which they were derived. Describing each in turn:

**A. Use conventional graphical objects where possible.** This directly reflects the theme "Unexplained symbols" (IE3). Participants expressed unfamiliar symbols as being confusing "maybe here those colourful images would be nice for someone who is familiar. In my case, I'm not really understanding them." (P3). The idea is also to positively impact two other themes: Consistency across diagrams (IE1) and Consistency within diagrams (IE2). This guideline is placed at the top of the list as it can have widespread layout impact, so would practically be useful to apply first.

**B. Only use one type of arrow for information flow.** This is to increase "consistency within diagrams" (IE2). It aims to address confusions such as "these little red arrows are not adding a lot" (P12) and "these arrows here are kind of a bit narrow. Obviously the fact it is bidirectional arrow is important but the arrowheads are extremely small." (P1).

**C. Use precision with care.** Directly related to the theme of "precision meaningfulness" (IP6), this aims to reduce unintentional over-specificity, as observed by the difference in interpretation seen in Table 3.

**D. Include the input and output of the whole system.** This is related to themes of "Diagrams as a Summary" (IW1), the use of "Diagram relation to the text" (IW4) and "Consistency within diagram" (IV3). The aim of this guideline is to alert authors to the readership use of the diagram for instantiation instantiation, and to encourage authors to consider using an

**Table 6. Proposed framework for improving neural network system architecture diagrams, as presented to participants during evaluation.**

| Guideline | Explanation |
|---|---|
| A. Use conventional graphical objects where possible | These are aesthetically preferred, and less likely to cause confusion |
| B. Only use one type of arrow for information flow | This is less likely to cause confusion. Reserve different types of arrow for fundamentally different uses |
| C. Use precision with care | Using (for example) 4 of a thing will make some readers think there are 4 of the thing and others $n$ of the thing |
| D. Include the input and output of the whole system | This helps make the overall purpose of the system clear |
| E. Consider using a single consistent example throughout | This helps some readers to understand by instantiating the example and then generalising |
| F. Use visual encodings meaningfully | When using a visual encoding principle, such as grouping by proximity or alignment, there should be a reason for it |
| G. Make navigation easy | Ensure it is easy to navigate a path through the diagram. Labels for layers, arrows, and linear alignment help to make navigation straightforward |
| H. Do not use colour for aesthetics | If you use colour, it should indicate grouping, otherwise it can cause confusion |
| I. Use available conventions | For example, if representing a CNN, it seems good to use the conventional 3D CNN format, and include all the filter widths numerically |
| J. Consider what people might expect to see | For example, if representing a CNN, put pooling in as a step. If you don't use pooling and that is important, consider noting that in a caption or label, as otherwise it may be assumed present |
| K. Be specific | For example, "BERT" is better than "embedding". This aids interpretability by avoiding obvious gaps |
| L. Consider that some readers may use the diagram without text | For these readers, a relatively self-contained diagram is particularly helpful |

**Table 7. Evidence from the interview study supporting each of the proposed guidelines in the framework.**

| Guideline | Interview Evidence |
|---|---|
| A. Use conventional graphical objects where possible | IE1, IE2, IE3 |
| B. Only use one type of arrow for information flow | IE2 |
| C. Use precision with care | IP6 |
| D. Include the input and output of the whole system | IW1, IW4, IV3 |
| E. Consider using a single consistent example throughout | IP2, IV3, IV4, IE1 |
| F. Use visual encodings meaningfully | IP5, IP6, IP7 |
| G. Make navigation easy | IP4, IV1 |
| H. Do not use colour for aesthetics | IP5 |
| I. Use available conventions | IE1 |
| J. Consider what people might expect to see | IW4, IP3, IE1, IE3 |
| K. Be specific | IW1 |
| L. Consider that some readers may use the diagram without text | IW2, IW4, IP1 |

example throughout if appropriate. By focusing on input and output, the guideline attempts to encourage making the purpose of the system clear. It also ensures that the scope of the diagram includes the whole system, making it more likely the diagram will be usable as a summary.

**E. Consider using a single consistent example throughout.** Related to the previous guideline, hence it's position, this aims to allow reasoning using the example (satisfying needs

expressed in IP2), but without prescribing diagramming details which could prevent the diagram from being "schematic" (IP1). Using throughout would also be "consistent within diagram" (IE1) and by being consistent, would also hopefully avoid "unexplained symbols" (as perhaps the example would be used instead) (IE3).

**F. Use visual encodings meaningfully.** Primarily, this is about "Ineffective use of diagrams relational properties" (IP5) - where visual encodings are used without meaning, it can cause confusion: "So it annoys me a little bit because it is making me think that those things are different while probably they are not." (P12). It is also necessarily related to "Precision meaningfulness" (IP6). This is especially relevant to the "Hyperparameters" (IP7) theme, where large variations in dimension size might be more effectively encoded by labels than by visual encoding such as size or repetition.

**G. Make navigation easy.** Directly addressing "Navigation" (IP4) and the expressed requirement for "Clear navigation" (IV1).

**H. Do not use colour for aesthetics.** This guideline encodes perhaps the most common violation of "Ineffective use of diagrams relational properties (IP5)". e.g. "I don't really know what the colouring of the arrows represents, if anything" (P1), "Why are these white then become yellow? The colour seems a bit arbitrary." (P11) and "Why some operations are grey, why some operations are blue, some operations are green. It is confusing for me. If they were all the same colour it would be easier for me." (P12).

**I. Use available conventions.** This is to aid "Consistency across diagrams" (IE1). "[Redrawing the diagram in a conventional style] is what I was talking about as a unified approach to doing these diagrams. In actual fact, on my PhD thesis one of the corrections they asked me to do was to redo the diagrams in this style." (P11).

**J. Consider what people might expect to see.** This relates to "Technical knowledge and familiarity" (IP3), and "Unexplained symbols" (IE3), and combined with Guideline K is intended to encourage appropriately complete and specific technical information. This guideline also aims to improve the community's "Consistency across diagrams" (IE1). It may also impact "Diagram relation to the text" (IW4), if authors are wanting to reference aspects of the diagram in text (or vice versa). It was felt unreasonable to use this towards "Diagrams as a summary", as authors may not be aware of this use case, hence separating out Guideline L.

**K. Be specific.** Since diagrams are sometimes used as a Summary (IW1), being specific avoids reference to text. This also came out as part of "Technical knowledge and familiarity" (IP3), in that if mentioned specifically in examples, it decreased ambiguity about how the system worked e.g. "Dense layer is very canonical, but the embedding layer could be anything, could be BERT, could be anything. It doesn't give me anything there." (P7).

**L. Consider that some readers may use the diagram without text.** Directly related to the previous, but more general, this aims to encourage thought about prioritisation of "Schematic overview versus implementation details" (IP1). This is also related to the theme that "Diagrams are perceived as effective compared with text (for relational communication)" (IW2), and to the theme "Diagram relation to the text" (IW4).

**4.11.8 Grounding in the interview study**

**4.11.9 Proposed framework in the context of literature** Whilst the proposed framework was derived from interviews rather than literature, one can also position the individual guidelines within literature. Some of the conceptually related work (e.g. around bias) are still debated topics, and indeed whether existing research would hold in this specific domain is unclear. However, aspects may be related and are outlined here. The individual guidelines are not independent of each other, and where there is overlapping related work this is signposted below.

**A. Use conventional graphical objects where possible.** This guideline was motivated by the confusion caused by the multitude of different graphical objects (visual components in the language of VisDNA) observed as being used to represent tensors. Physics of Notations advocates "graphic economy" [115], keeping the number of different graphical symbols cognitively manageable. Physics of Notations is designed for defined visual notations, rather than heterogeneous diagrams, but the concept of graphic economy over this corpus (and within a single diagram) may still be relevant. This guideline can also be viewed as a specific case of guidelines I and J.

**B. Only use one type of arrow for information flow.** The importance of arrows in showing functional information in mechanical diagrams has been shown [63]. Schneiderman [112] advocates internal consistency. Physics of Notations' "semiotic clarity" advises a "1:1 correspondence between semantic constructs and graphical symbols" [116]. This guideline intends to support this principle.

**C. Use precision with care.** Concreteness may impact the ability to transfer conceptual knowledge, as found in children's mathematical ability [117]. Precision, or *unintentional concreteness*, may impact this negatively. Particularly when combined with "confirmation bias", that people choose to search for the information that confirms their own hypothesis [118], unintentional concreteness or meaningless precision may facilitate readers in drawing incorrect conclusions. See also guideline E, as a more general case of instantiation.

**D. Include the input and output of the whole system.** Bäuerle et al [25] recommend that CNN visualisations include input and output samples, but did not integrate this with the Net2Vis technical solution for practical reasons, providing a placeholder instead. See also guideline E for discussion of instantiation, as often the input and output are instantiated (though this is not recommended by this guideline itself).

**E. Consider using a single consistent example throughout.** In children, "Concreteness may have some advantage over generic for learning" [117]. However, concreteness may also limit the ability to abstract to general principles and to transfer knowledge to different settings, leading some educators to advise avoiding concrete instantiations [119]. Still others suggest instantiations are a useful part of an educator's toolbox [120]. In this case it is particularly unclear how the related pedagogical research would correspond to experienced scholarly users.

**F. Use visual encodings meaningfully.** Tufte [45] can be seen to support this, as he states the relations between things is crucial. Gurr [121] describes how mis-use of visual encodings such as misalignment can lead to unwanted implications.

**G. Make navigation easy.** Cheng [122] suggests that minimal paths makes diagramming sometimes more efficient for answering questions than other representations such as sentences and tables. In his recent "pattern language" to support creating new diagrams, Blackwell [62] notes that easy navigation may reduce reliance on working memory.

**H. Do not use colour for aesthetics.** Somewhat, the idea of data-ink could be said to support this [45]. Similarly, colour palette choice has been shown to have different implications for different users [123]. However, colour is important in many measures of aesthetics (see [124]), and aesthetically appealing diagrams may be more effective at communicating and having impact. For example, Pauwels comments that "Trying to exclude 'aesthetics' from science is an illusion" [125]. This guideline is also related to guideline F, in the case that colour is used in such a way as it may be interpreted as a meaningful visual encoding (of grouping, for example).

**I. Use available conventions.** Social semiotic aspects may be improved by utilisation of visualisation conventions, which do "persuasive work" for example by giving an "aura of objectivity" [126]. Further, one can argue that Schneiderman's [112] advocacy for internal

consistency may be generalised to contextual consistency. In terms of individual visual components, fulfilling expectations (e.g. using expected glyphs) has been shown to facilitate object recognition [127].

**J. Consider what people might expect to see.** In visual cognition, "Fulfilled expectations are associated with facilitated object recognition but attenuated neural responses". In scholarly communication of physics, Rowley [39] notes the visual aspects of conference slides, and considers visual communication of this nature to be a social semiotic, including a social dimension. Rowley comments that meeting visual expectations aids following an argument, and that the absence of expected visuals will be spotted by experts as an anomaly, and weaken claims.

**K. Be specific.** Bäuerle et al [25] suggest including layer data, whilst providing flexibility in Net2Vis to aggregate. The questions they use to evaluate the suitability of Net2Vis are very specific, suggesting the authors value specific information in scholarly NN diagrams. Specificity (to an expected or conventional level) is also related to social aspects discussed with respect to guideline J.

**L. Consider that some readers may use the diagram without text.** In examining different visualisations of scientific facts, including photographs and comic strips, Walsh et al. [128] found that most participants preferred a cartoon style over text alone, and conclude that short term recall of scientific facts may be best done by providing different options to support reader preference for visual accompaniment to text.

### 4.11.10 Limitations

- **Limitation:** The framework is primarily induced from interviews, and may over-fit to the participants. **Comment:** Empirical evaluation will be conducted from several different angles in Chapter 7, which reduces the risk that the framework will be only of use to the interview study participants. The linkage with literature further reduces this risk.
- **Limitation:** The framework has not been quantitatively grounded in order to determine priority. **Comment:** The framework is evaluated as a whole and as individual guidelines. Whilst it may be that unimportant items are included, these will be highlighted in the evaluation and recommendations for changes made. Further, the framework is not designed to be an end-point: As discussed in Sect 4.11.1, the framework could be amended as representational challenges emerge and as user requirements change.
- **Limitation:** By not being a fundamental part of the publishing process, the framework risks having limited adoption and impact. **Comment:** The evidence-based evaluation allows for impact in many application areas, including as part of adoption within the publishing process. Further, proposals for tutorials and workshops at NN venues have been submitted to improve visibility. Pragmatically, a soft-launch was felt more sensible than engaging with publishers or tool creators directly at this stage. Further, this was useful for constraining the thesis scope.
- **Limitation:** By not being based on existing design principles, the framework risks being specific, and only useful to this domain. **Comment:** The framework is intentionally specified for the domain of scholarly NN system diagrams. Not basing on existing principles allows priority to be given to addressing the most urgent issues, and allows a more intuitive combination of design and content issues which is intended to be more natural for NN system scholars, instead of using language and concepts which may be unfamiliar and a poor cognitive match for potential framework users.

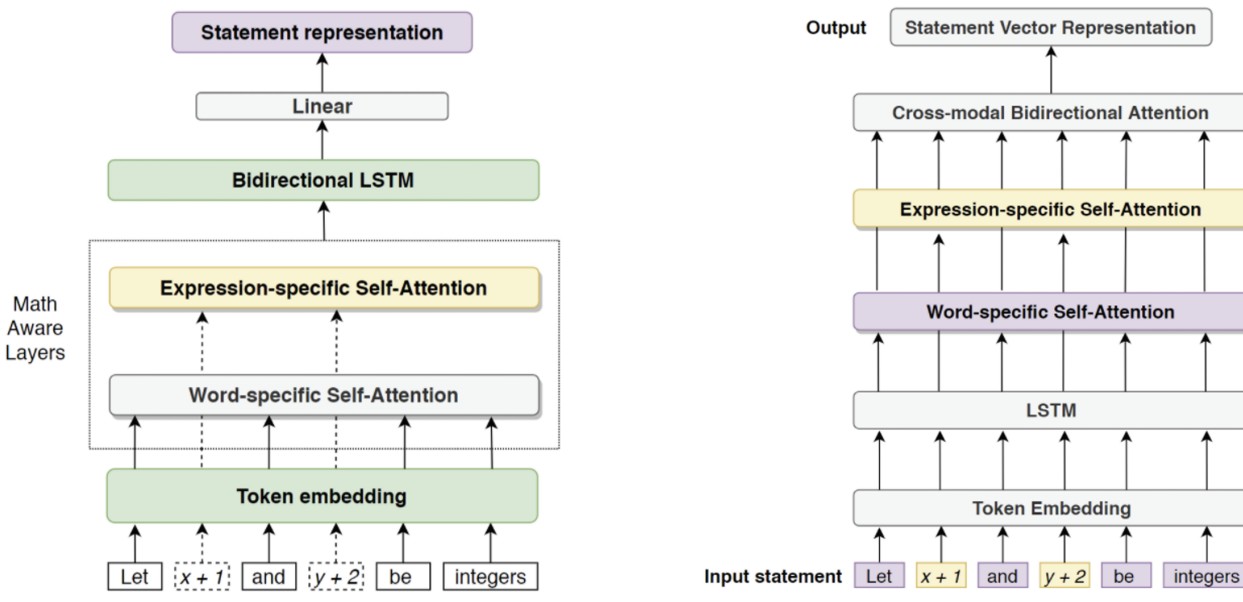

**Fig 6. B1P1.** "Before framework" = left, "After framework" = right.

## 5 Results II: Framework evaluation

### 5.1 Results

The results are reported by aggregating both groups together, except where specified.

**5.1.1 Diagrams created during the study** The following figures document some of the diagrams used. The full collection of diagrams is available [129]. In Figs 6, 7, and 8, the original diagram is above or on the left, and after exposure to the framework is below or on the right. Labels are not used within figures in order to avoid confusion with the diagrams themselves.

Fig 6 has been edited to add explicit output, has labelled input and output, changed the colouring, removed a grouping box, added more specifics (e.g. "Cross-modal Bidirectional Attention" and the LSTM layer) and added many more arrows. Whilst the multiple arrow addition is perhaps not economical with ink, it does make explicit that each input is processed through to the final layer - which was perhaps not obvious from the "before framework" diagram.

Fig 7 includes input and output, adds ellipses, and makes the arrow directions more consistent. It also includes a legend, beyond the framework's advice but perhaps helpful.

Fig 8 removes the "graph" and adds much more detail about the system itself (e.g. "feature map", "soft-max"). The output is also made explicit. A new convolutional layer graphic is included, and the relation between "recursive feedback" and the sentence generation is made clearer. Note that multiple arrow types continue to be used. As with Fig 7, the essence of the framework has been applied, with deviations.

**5.1.2 Initial reported participant opinion on which guidelines to keep or remove** Table 8 shows the count of recommendations to keep each guideline. Most were generally agreed as useful on first inspection by authors. B, E, H, and L were the least recommended. Free text was available, but no participants added comments about why they would not recommend specific guidelines. This was after the first authoring, i.e. participants had not seen

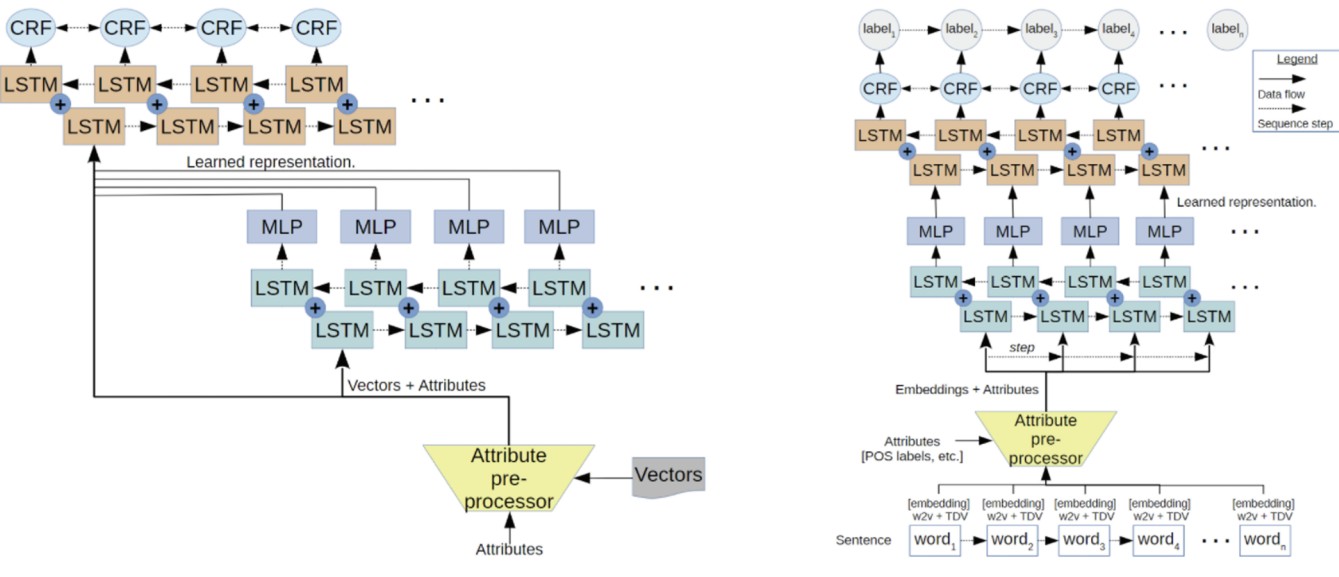

**Fig 7. B1P3.** "Before framework" = left, "After framework" = right.

any other diagrams at this stage, and the data reflects author views (or *expected* readership views) rather than *actual* readership views.

As readers, comments sometimes tangentially referred to the framework, such as "I prefer [B1P5 post-framework], since the use of BERT is more clear [sic]." (B1P1). Referring to B2P4, B2P3 said "ab[b]reviations should be explained". This may reflect different NN specialisms of participants, but also points towards the idea of making the diagram fairly complete (or at least usable without supporting text).

**5.1.3 Reported participant opinion on the framework overall**  The final views of participants having undertaken the full experiment are collected below, and provide qualitative evidence of the utility of the framework. 9/10 participants recommended using the framework, providing qualitative evidence supporting the framework. Participants had the following final comments:

**Table 8. Count of participants recommending to keep each guideline. Most participants thought most individual guidelines should be kept.**

| Guideline | Count of participants recommending to keep |
|---|---|
| A | 10 |
| B | 7 |
| C | 9 |
| D | 9 |
| E | 7 |
| F | 10 |
| G | 10 |
| H | 8 |
| I | 9 |
| J | 10 |
| K | 9 |
| L | 8 |

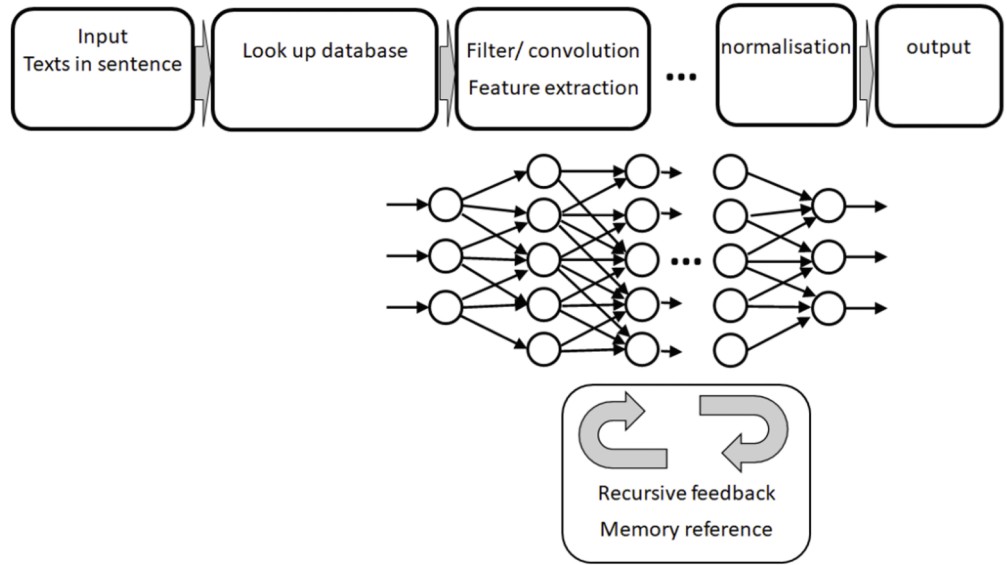

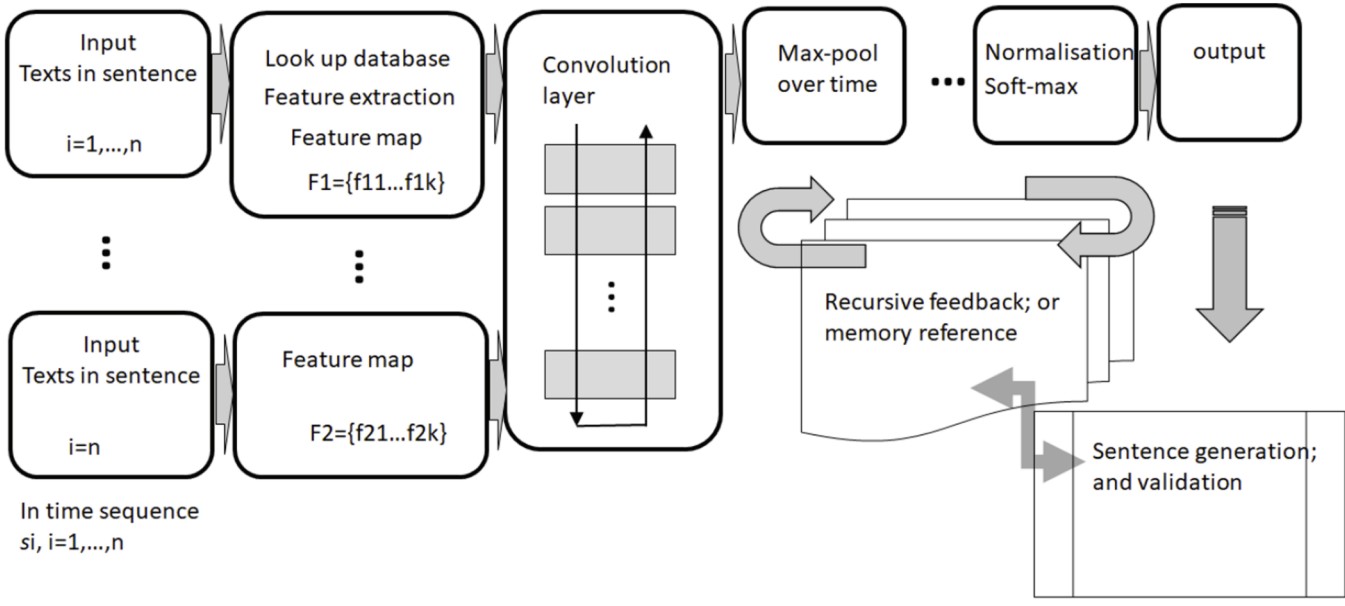

**Fig 8. B2P2.** "Before framework" = above, "After framework" = below.

- "Overall the guidelines provide an excellent standard to follow, and I am sure that lots of my peers would benefit from that." (B1P1)
- "I think especially for people who are new to the construction of Deep Neural network diagrams, these guidelines can help them illustrate their work better." (B1P2)

- "As in textual communication, the use of common expressions and jargon among professional groups has a tendency to facilitate and accelerate the diffusion of knowledge. The guidelines can be seen as an attempt to apply the same concept to visual language." (B1P3)
- "They make the basis of a checklist of good design practice, where I think a lot of design decisions are made somewhat unthinkingly." (B1P4)
- "Neural network diagrams are capable of conveying a very complicated approach succinctly. As a reader, it is the first thing I try to read in a paper. However, currently, there is no uniformity in the way the diagrams are presented. I believe this would be a good starting point in reaching that." (B1P5)
- "That would make the problem more clear [sic]" (B2P1)
- "The instruction and information are too ambiguous to proceed the tasks. [sic]" (B2P2) (This participant did not recommend the framework)
- "[H]elpful to describe complex networks" (B2P3)
- "I think they are a very sensible set of checks to perform on your neural network / system diagrams." (B2P4)
- "As ML researchers, we are taught very little about how to write, how to draw, how to propagate our ideas. I think these guidelines make this job easier for us and should be publicized." (B2P5)

B1P5's comment links to the finding from the interview study, that some scholars are reading the diagram before text (IW4 in Sect 4.1). B2P5's comment links to the introductory motivation commentary about the lack of support for scholarly diagramming (Sect 1.7).

**5.1.4 Utility of the framework for increasing reported preference** The study was not designed to show statistical significance, especially with the low number of participants. A chi squared test was employed to investigate whether total preference was significantly higher in post-framework diagrams than in pre-framework diagrams, but this did not produce a statistically significant result. Other relationships *not* involving preference did produce statistical significance and are reported in "Other Results".

Table 9 shows preference of each pair of diagrams. This statistical analysis was done in Excel. Of the edited diagrams, participants rated the post-framework diagram as preferred after editing 50% of the time. They were seen as equivalent to the original in 14% of cases, and the original version was preferred in 36% of cases. By examining the diagrams (especially B1P1 and B2P5, which are negative changes in terms of preference), it appears that part of the previous version preference may be due to unintentional interpretation of the arrow guideline by authors, which may have led to changes that were not preferred by authors (and were not intended to be prompted by the framework). In terms of diagram preference, B1P2 divided opinions, as did B2P2. In B1P2, the two main changes were arrows (qualitatively negative) and adding system output (qualitatively positive). It may be that for readers the priority given to either of these altered their opinion. In B2P2, the diagram underwent major changes and identifying possible reasons for the polarised views between versions is challenging.

The slight positive effect of the framework measured quantitatively is also reflected in qualitative feedback:

*…the concrete changes to the diagrams were small, but at the same time, the feedback highlighted what the study is seeking to solve. With both information (the diagrams and feedbacks*[sic]*) I was quite convinced of the relevance of this study and the applicability of its resulting guidelines.* (B1P3)

**5.1.5 Usability** Most participants (7/10) chose to edit their diagram when exposed to the framework, suggesting the framework is usable. However, 3/10 participants did not edit their

diagram, which could be interpreted as poor framework usability. This lack of editing may be a feature of the experiment's design, as the authors may not have had time to edit a diagram which (in some cases) was already published. In some cases, the guidelines were not followed. For example, Figures 6, 7, and 8 include multiple types of arrows. This may suggest poor usability, or that these specific guidelines were not seen as useful by participants. However, there was a lack of negative qualitative comments on this topic. See Limitations in Sect 3.3.2 for further discussion.

That most individual guidelines were recommended by most participants suggests they saw the overall framework as useful. 9/10 participants gave positive feedback about the experimental method. One participant made comments suggesting that the framework was difficult to use, and that the experiment design was confusing: "More clear [sic] and specific instructions for the experiment and consistent use of terminology could be considered." (B2P4). B1P3 disagreed with the guideline about one type of arrow, but added a legend to clarify their visual encoding choice, which is within the intention for the framework.

**5.1.6 Other results** The main others results are:

- Author rated "correctness" and "confidence" in reader text answers are correlated, suggesting it may economise to only request an overall score.
- An overall highly-scoring diagram according to readers is associated with overall good reader text summary, according to the diagram author (a chi-squared test using R gave p = 0.003). See Fig 9. This perhaps suggests that readers have a reasonable grasp over what they do not understand, or that simple diagrams are easier to understand correctly. However, manual inspection of the diagrams suggests that the "simpler diagrams" hypothesis is likely to be insufficient to explain this.
- Reader diagram quality is associated (statistically significantly) with author assessment of "overall score" and "completeness", but not "correctness", of the reader's textual answer.

Fig 9 shows that good diagrams (as perceived by readers) is correlated with those readers providing a good overall text summary (according to the diagram author). The outliers in Fig 9 are primarily due to four responses to Batch 2 P4 which were *all* "I do not understand", which the author scored positively as good answers. These have been included in the analysis as the study method is designed to avoid placing quality-assessment into the hands of the study administrator. There was no statistically significant relationship found between reader

**Table 9. Count of participants preferring each version of the diagram. Diagrams B1P4, B2P3 and B2P4 were not edited.**

| Diagram ID | Post-framework preferred | Neither preferred (Null) | Pre-framework preferred |
|---|---|---|---|
| B1P1 | 1 | 0 | 3 |
| B1P2 | 2 | 0 | 2 |
| B1P3 | 2 | 1 | 1 |
| B1P4 | 4 | 0 | 0 |
| B1P5 | 3 | 1 | 0 |
| B2P1 | 0 | 4 | 0 |
| B2P2 | 2 | 0 | 2 |
| B2P3 | 0 | 4 | 0 |
| B2P4 | 0 | 4 | 0 |
| B2P5 | 0 | 2 | 2 |
| Total | 14 | 16 | 10 |

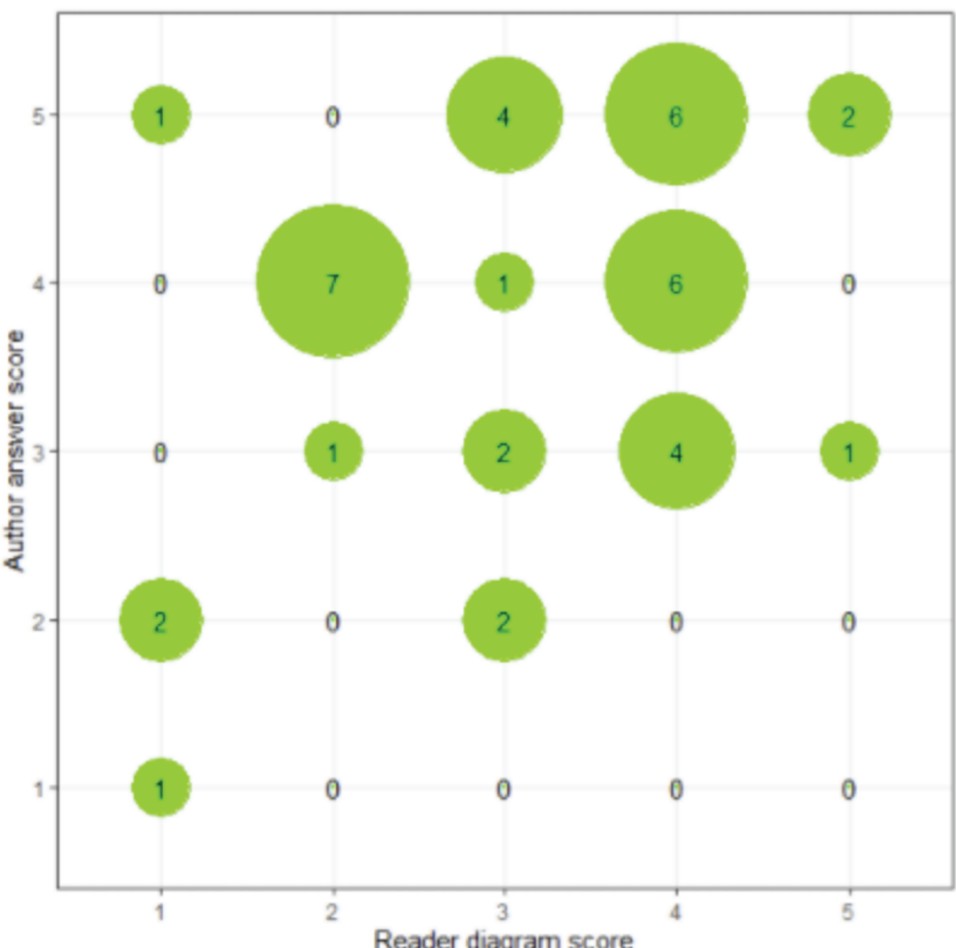

**Fig 9. Good diagrams as perceived by readers is correlated with those readers providing a good overall text summary according to the diagram author.**

confidence and author score. The results suggest it may be sufficient to consult readers alone, for improved diagrams.

**5.1.7 Reflections on the methodology**  9/10 participants felt the study was worthwhile, leaving reflections such as "Since I believe this to be a first attempt at such a proposal, the concrete changes to the diagrams were small, but at the same time, the feedback highlighted what the study is seeking to solve. With both information (the diagrams and feedbacks [sic]) I was quite convinced of the relevance of this study and the applicability of its resulting guidelines." (B1P3).

Two participants noted the limitation of considering the diagram in isolation: "I would not think that we can make an appropriate judgement only with the diagrams. Depending on what it is for, the system should be very different. It is in fact very dangerous to assess the diagram without any instruction or description of the system, if it is for a research publication or conference presentation." (B2P2) and "one thing that has been a little odd is the lack of context from viewing diagrams in isolation. I think this has made certain questions a little hard to answer." (B1P4).

One participant (B2P3) commented that "5 participants is low". Whilst this may be a reasonable comment, the participant may also not have been aware of the wider context of their participation (10 participants). The experiment was originally designed as a workshop series for many more participants. The digital administration and broader the COVID-19 context for participants is likely to have contributed to challenges in recruitment and resulted in a lower-than-desired number of participants for this study. Despite this, it is hoped that the reader can see the value in this small-scale study, particularly when combined with the subsequent corpus analysis.

The effect of diagram ordering was also analysed, and found to make no significant difference to reported preference.

**5.1.8 Future study effect-size** It is possible to estimate the sample size that would be required to achieve statistical significance for similar studies in future. Doing this for the key hypothesis that "post-framework will be preferred", it is assumed that the data would be distributed according to what was gathered with the 10 participants. For simplicity, we consider only the cases where there was non-neutral preference (23 cases). Of these, 14 participants preferred post-framework and 9 preferred pre-framework versions. This can be compared against the null hypothesis (that both pre- and post- framework are equally likely to be chosen).

From Binomial tables we can find the minimum value of the test statistic. Assuming $X \sim Binomial(n, 1/2)$, then we want the smallest $N$ such that $\mathbb{P}_n(X \geq N) \leq 0.05$. These are shown in the first two columns of Table 10. Assuming that the test statistic $Y \sim Binomial(n, 14/23)$ we can then calculate the probability that our experiment would reject the null hypothesis, shown in the third column of Table 10.

This suggests that a total of 230 diagrams would be required to have a high probability of being able to reject the null hypothesis and establishing statistical significance.

## 5.2 Summary

This analysis suggests the *usability* of the framework is satisfactory but may benefit from improvement, and the *utility* of the framework is reasonably high. Utility has been demonstrated by (i) higher reported preference (ii) author-centric feedback stating the framework was useful (iii) reader-centric feedback stating the framework was useful. Likely effect-size for similar future studies was also established.

The next section triangulates the findings, examining ACL 2017 diagrams and their compliance with the framework, further demonstrating the utility of the framework. The union

**Table 10. Cumulative probabilities, for diagram sample size (n) and simulated number of post-framework preferred N, with probability _p_ of being a sufficient sample size to demonstrate significance.**

| Non-neutral diagram assessments (n) | Simulated number of post-framework preference (N) | $p = \mathbb{P}(Y \geq N)$ |
|---|---|---|
| 23 | 16 | 0.264393 |
| 50 | 32 | 0.382638 |
| 100 | 59 | 0.688255 |
| 200 | 113 | 0.909064 |
| 229 | 128 | 0.945575 |
| 230 | 128 | 0.953627 |
| 500 | 269 | 0.99944 |
| 1000 | 527 | 1.000000 |

of these studies suggests that the framework does indeed facilitate improvement of NN diagrams.

# 6 Results III: Corpus analysis

## 6.1 Introduction

In addition to the empirical study reported above, the *utility* of the framework is examined using an alternative "corpus-based" approach. The term "compliance" is used here to mean the extent to which each diagram obeys or violates the guidelines, as a quantitative measure: $1 - \frac{\text{number of guidelines violated}}{\text{total number of guidelines}}$. The analysis in this Section attempts to use compliance with the guidelines as an indicator of diagram quality. It also shows that compliance with the framework is correlated with higher citation counts at ACL 2017 after 3 years. In addition to the present interpretation of the metric, citation prediction is also a scholarly activity in its own right, said to be useful for "guiding funding allocations, recruitment decisions, and rewards" [130].

This subsection is based on an extended version of [4], and reports the parts related to the relationship of the framework with the ACL 2017 diagram corpus.

## 6.2 Method

This analysis follows the same overall method as reported in Marshall et al. [4], using the accompanying dataset [18]. Briefly, publication metadata was manually extracted from all long papers from ACL 2017, including number of citations. In each paper, at most one extracted diagram was identified as the primary system diagram.

For the experiment presented here, additional metadata was added, including conformity to each individual guideline, and whether the diagram was colour or monochrome. Inter-rater reliability was measured on a subset of diagrams to validate scoring of framework compliance, with Gwet's $AC_1$ coefficient finding "good" reliability, and subsequent assessment was done with a single coder. The quantitative corpus data was exported as a csv, and the statistical package R was used for this analysis.

## 6.3 Results

Recall that 119 of 124 system diagrams at ACL 2017 described neural network systems (the others being diagrams of an embedding only, or not a neural system). These 119 diagrams were assessed against each of the 12 guidelines established following the interview study, which are reproduced as part of Table 11. These guidelines were chosen in favour of other diagramming guidelines due to their domain specificity.

In addition to exploratory analysis, this study aimed to test the hypothesis that fewer diagram guideline violations would be observed in papers with a higher number of citations. In this analysis, a correlation was found between number of citations and "specific" ($p<0.05$), and also "self contained" ($p<0.05$) guidelines. The other guidelines alone did not correlate with a significant difference in number of citations. However, the best correlation was found with the average of guideline compliance, rather than any individual guideline. The results of this are shown in Fig 10. Papers with diagrams conforming with 10/12 guidelines or greater are likely to have a higher number of citations than those conforming to fewer than 10 guidelines.

LOESS (locally estimated scatterplot smoothing) is a technique for moving averages for scatterplots, using moving average and polynomial regression. The LOESS curve in Fig 10 can be approximated by an exponential function, $citations = e^{10(compliance - 7/12)} + 14$, where "7/12"

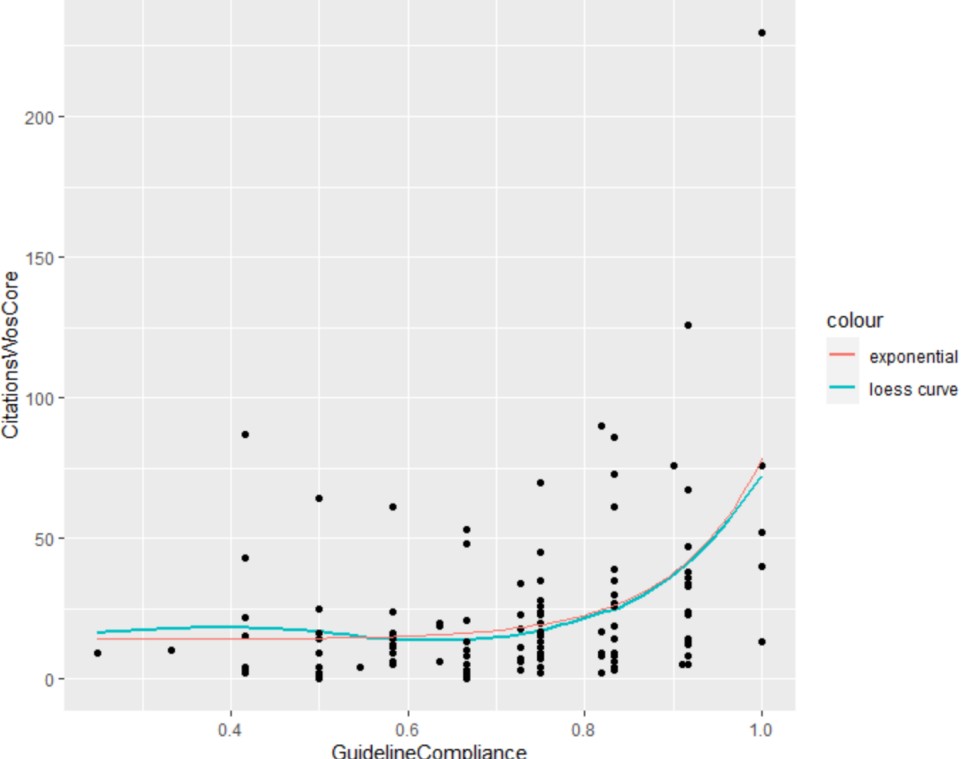

**Fig 10. Scatter plot of number of citations versus NN system diagram guideline compliance (as a quantitative proxy for "how good the diagram is").** LOESS curve for locally weighted smoothing is in blue, and the function $y = e^{10(x-7/12)} + 14$ is in red.

encodes the increase in citations observed from higher levels of compliance, "14" encodes the asymptotic average number of citations for low compliance papers, and the multiplier "10" fits the curve. The only independent variable is average guideline compliance in each diagram. Note that LOESS smoothing can overfit due to the sensitivity of the model to parameters. This

**Table 11. Framework violation count, for each quartile of number of citations.**

| "Number of citations" quartile | Bottom (30 papers) | 2nd (31) | 3rd (28) | Top (30) |
|---|---|---|---|---|
| No prevalent unconventional objects | 3 | 3 | 2 | 5 |
| One arrow for information | 17 | 17 | 17 | 14 |
| Precision care | 15 | 11 | 8 | 15 |
| Input and output | 2 | 4 | 4 | 4 |
| Example | 18 | 16 | 10 | 11 |
| Meaningful visual encoding | 10 | 7 | 8 | 3 |
| Easy navigation | 7 | 8 | 7 | 4 |
| Colours not aesthetic only | 12 | 13 | 16 | 8 |
| Conventions | 12 | 8 | 7 | 4 |
| Expectation matching | 6 | 5 | 7 | 1 |
| Specific | 17 | 11 | 15 | 5 |
| Self contained | 6 | 6 | 5 | 0 |

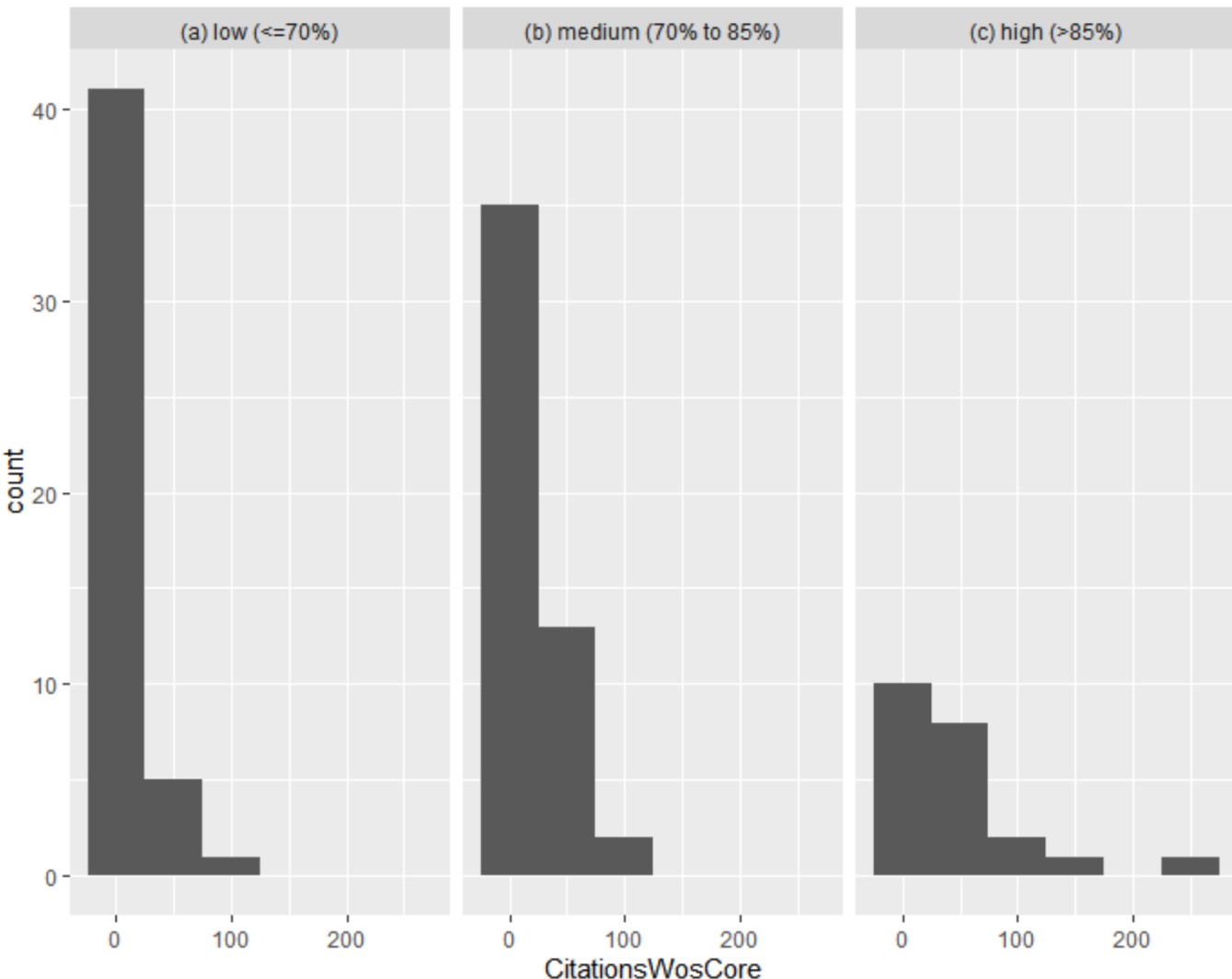

**Fig 11. Number of citations for papers containing system diagrams, grouped by level of guideline compliance.**

work does not aim to model citations accurately, but to argue that the framework captures some diagramming behaviours of effective communicators.

For diagrams conforming to fewer than eight guidelines, it appears visually almost to be random whether the author has conformed to the framework or not. Fig 11 shows diagrams with high, medium and low compliance (the percentage boundaries were chosen to make the population sizes similar). This simplifies the results of Fig 10, and shows that papers containing diagrams complying with over 85% of guidelines are less likely to have a lower number of citations, and more likely to have a higher number of citations.

Of diagrams conforming to 11/12 guidelines, six used more than one arrow, two did not use examples, one used unconventional objects, and one violated the colour guideline. The reason for the multiple arrow types in each case was often evident by inspection, usually to

separate a type of data flow or distinguish between abstraction levels (e.g. mathematical workings of a neuron vs data pipeline). It may be advisable that the "one type of arrow" guideline should be revisited through user evaluation. The authors are evidently good communicators, in terms of having above-average number of citations, and this guideline not being observed by those authors supports the view that neural network system diagrams may be better supported by a flexible framework rather than rigid standards [131].

In an attempt to identify whether diagrams were formed using good diagramming practices, papers in the top quartile of citations and including a NN system diagram (of the 119 NN system diagram papers, this includes the 30 papers with 28 or more citations) were examined for their conformity to the framework. "Not applicable" scores were omitted from the analysis. Table 11 shows the results. No top quartile paper violated the self-contained guideline, meaning those diagrams were understandable on their own and did not reference text directly. Including input and output, which is often related to self-containedness, was also done by all except three of the top 27 papers. Further, whilst avoiding unconventional objects was done in 25/30 top quartile diagrams, this perhaps came at some cost of using precision with care, where that is conventional (12/30 not conforming to this). Almost half (14/30) of top quartile papers used multiple arrow types, suggesting that thoughtful use of multiple arrows may be good for communicating different abstraction levels. Using an example in the diagram was done in 19/30 highly cited papers, and those that did not include an instantiated example often included mathematical notation (11/19).

Of the 119 NN system diagrams, 64 diagrams (54%) contained an explicit example to instantiate the input. If an example was used, it was almost always used in the input, and often in the output, and occasionally at intermediate steps. There was not a significant difference in citations whether the paper's system diagram included an example or not, and both categories appear to follow a similar distribution, though the most highly cited papers included an instantiated example.

Of the 30 top-cited papers, nine did not use colour, 13 used colour meaningfully, and eight used colour for aesthetics only. This suggests that colour for aesthetics may be appropriate in some cases (or it may be emphasising aspects of the system in a manner not uncovered by the present method). However, examining relative frequency of occurrence, it is found that highly cited papers are less likely to use colours for aesthetics only.

Framework compliance was not correlated with abstract including an "architecture" keyword, nor with conference area, suggesting that the framework is equally applicable to each particular contribution type. This corpus analysis suggests that the framework may be generalisable to cases beyond the interview study and experimental framework evaluation.

## 7 Results IV: Quantitative insight from Design and Domain experts

### 7.1 Introduction

To gain insight into whether the framework is encoding only good design practices, three information Design experts (a Senior UX Researcher, a Graphical Designer, and a Product Design Consultant) shared their insight on design.

### 7.2 Method

- Three information Design experts (a Senior UX Researcher, a Graphical Designer, and a Product Design Consultant) and three neural network domain experts (all PhD Computer Science candidates) were recruited via personal contacts of the researcher.

**Table 12. Correctness in partitioning top and bottom cited papers based on diagram only. Framework compliance outperforms the best performing expert assessment, in addition to consensus-based measures.**

| Predictor | Correctness |
|---|---|
| Design expert 1 | 50% |
| Design expert 2 | 50% |
| Design expert 3 | 50% |
| Design expert consensus | 50% |
| NN expert 1 | 70% |
| NN expert 2 | 60% |
| NN expert 3 | 60% |
| NN expert consensus | 60% |
| Framework compliance model | 80% |

- The ACL 2017 diagram corpus [18] was used to extract the highest- and lowest- cited 10 diagrams from ACL 2017. Captions were included.
- The diagram order was randomised.
- Experts were sent these 20 diagrams, as a form attached to an email.
- The experts were asked to guess which group they expected each diagram to be in, with free text to explain their thoughts.
- The results were collated, and the modal guess for each expert type was calculated.
- The framework compliance (included in the diagram corpus data) was used as a "classifier" to put the diagrams into two groups of high and low framework compliance.

## 7.3 Results

Table 12 shows the results. At an individual level, all three Design experts correctly guessed 50% of diagrams. Note that 50% is the expected score if selecting responses at random. Taking the modal response from the three Design experts gives a "Design expert consensus" correctness of 50%.

No free text response was given by any NN Domain expert. Every Design expert provided a textual response, for most questions. The free text responses given by Design experts seem to suggest that complex diagram models were penalised by Design experts, which did not match the paper citations, particularly where navigation through the diagram was not linear. A lack of inputs and outputs were also mentioned, as was inconsistent use of colour, differing arrow types, and level of domain knowledge required. Some aspects mentioned by Design experts, such as confusing use of shading, were not included in the framework. Note that participants had not seen the framework.

A chi-squared test shows a significant difference in overall performance between Design and Domain experts ($p < 0.05$). The higher Domain expert score could be because these experts were familiar, consciously or subconsciously, with the paper or diagram. The findings suggest that domain knowledge should be considered, not just design principles, when creating scholarly diagrams.

A simple model of framework compliance, classifying into whether a diagram is above or below "average" framework compliance gives a 80% success rate at predicting whether the containing paper is top or bottom cited. This simple model outperforms any individual expert assessment, and also their consensus, and suggests the framework encodes a combination of both domain and design practices.

Combined with the corpus analysis and evaluation study, the expert study builds confidence in the usefulness of the guidelines, as these studies suggest the guidelines may (i)

include items that are relevant to the manifest phenomena (ii) be usable and reported as useful by authors and readers and (iii) capture something beyond expert-assessed design- or domain-only features.

# 8 Discussion

## 8.1 Methodology

The methods employed to evaluate the framework are twofold: (i) the study where participants are asked to edit their own diagrams and provide peer feedback, and (ii) the corpus-based evaluation. Combined, these approaches suggest (i) that exposure to the framework is useful to practitioners and (ii) that the framework could have a widespread utility to the scholarly community. Further, the results suggest that the framework encodes not just good design- or domain- practices, but a combination of the two. As such, it may provide a good basis for developing a more standardised diagrammatic language, supported by tooling, as the discipline and it's representational requirements continue to mature.

## 8.2 Incorporating evaluation results into the framework

Table 13 integrates the results of the evaluation, indicating that the guidelines B, C and I were least useful, and groups the guidelines into categories of "Visual Encoding" and "Content" for ease of use. These have then been ordered by least violations in the top cited quartile papers, in an attempt to further improve ease of use by having the most impactful guidelines at the top of each section. The lettering of the previous has been replaced by enumeration, to make salient this prioritisation.

**Table 13. The framework for improving neural network system architecture diagrams, incorporating evaluation results.**

| Category | Guideline | Explanation |
|---|---|---|
| Content | 1. Consider that some readers may use the diagram without text | For these readers, a relatively self-contained diagram is particularly helpful |
| | 2. Consider what people might expect to see | For example, if representing a CNN, put pooling in as a step. If you don't use pooling and that is important, consider noting that in a caption or label, as otherwise it may be assumed present |
| | 3. Include the input and output of the whole system | This helps make the overall purpose of the system clear |
| | 4. Be specific | For example, "BERT" is better than "embedding". This aids interpretability by avoiding obvious gaps |
| | 5. Consider using a single consistent example throughout | This helps some readers to understand by instantiating the example and then generalising |
| Visual Encoding | 6. Use visual encodings meaningfully | When using a visual encoding principle, such as grouping by proximity or alignment, there should be a reason for it |
| | 7. Make navigation easy | Ensure it is easy to navigate a path through the diagram. Labels for layers, arrows, and linear alignment help to make navigation straightforward |
| | 8. Use conventional graphical objects where possible | These are aesthetically preferred, and less likely to cause confusion |
| | 9. Do not use colour for aesthetics | If you use colour, it should indicate grouping, otherwise it can cause confusion |

## 8.3 Relation to Zobel's diagramming guidance

As discussed in Sect 1.7, few scholarly guides include guidance on diagrams, with the exception of "Writing for Computer Science" by Zobel [44] which includes advice relating to Computer Science diagramming. The results validate the majority of Zobel's opinions (relating to clutter, reducing ink, ). Zobel's comment that "schematic showing data flow in an architecture is likely to be unclear if control flow is also illustrated." reflects some of the representational tension observed in NN systems. However, an example Zobel explains at length that lines should not be overly thick, something which was not felt high priority in this NN study. A further difference is that our results suggest that it is not especially important that "boxes have different meanings in different places", though our "meaningful encoding" was felt un-empirically (but theoretically) important.

In summary, the present framework refines and provides additional scientific grounding to support Zobel's claims, and may be a supporting artefact for Zobel's suggested behaviour to "revise your pictures as often as you would your writing".

## 8.4 Practical implications and challenges of adopting this framework

As discussed in Sect 4.11.1, a major challenge with this framework is that it does not provide all the cognitive support of a fully-fledged diagramming language. There is no software tooling available with this approach which could aid adoption. The framework has been designed to be simple, and framework adoption is perhaps primarily about visibility in a fast-paced, highly saturated research environment. A further aid would be reviewers of papers examining diagrams more closely, avoiding errors in published diagrams [22]. Practically, implementing accessibility and visual design guidelines as part of the publication process would aid this. Further challenges, as outlined in Sect 1.7, are that new researchers are often not taught how to diagram effectively, so there isn't a pre-existing training delivery mechanism in most higher education institutions for this type of content. As the level of urgency around effective communication about AI rises, it is hoped this foundational work will help inform other initiatives around AI transparency and explainability, noting the importance of diagrams in expert-expert communication.

## 8.5 Future research

There are many possible avenues for research which could prove useful to the community:

- Extended framework evaluation could follow the same method at a larger scale. Other methods based on comparing interpretation of diagrams of the same system would also be beneficial.
- The neural network diagram corpus could be extended to different venues and over a longer duration, with statistical analysis replicated and additional comparison between venues.
- Investigating the propagation of diagramming techniques through the "social network" of papers would be useful for understanding how to make impactful changes to diagramming techniques.
- At a larger scale, having consistency across diagrams is important for cognitive efficiency and for widespread community adoption. Development of diagramming tooling, or using these guidelines to support the creation of a language supported by tooling, would be expected to have significant advantages in terms of adoption.

## 9 Conclusion

Diagrams are an important and widely used way of communicating the architecture of neural network systems. Our interview study finds heterogeneity in the way they are constructed and understood, which provides freedom for the author, but leads to potential inaccuracies in their interpretation. Existing HCI guidelines have relevance for scholarly neural network system diagrams, but no set maps directly to the issues we uncovered in the study. To bridge this gap, we propose a framework specifically addressing the main causes of confusion.

The framework was evaluated using a novel method for capturing both authorship and readership properties of diagrams, which measured the impact of the framework qualitatively and quantitatively. In addition to being recommended by participants in this study, the framework was demonstrated to have a small positive impact on diagrams created by authors. Further, in a corpus-based approach, high compliance of diagrams with the framework was found to be correlated with higher citation counts in the paper containing them. Finally, involving design and domain experts suggested the framework captures good diagramming practices beyond those identified by either design experts or domain experts alone.

Through three distinct evaluations, the utility of this framework is demonstrated, both in theory and in practice. We conclude with a participant comment that concisely summarises the findings of this study: *"I think this lack of language for diagrams is so bad, even at a high level there is nothing the same at all."* (P10).

## Acknowledgments

Thanks to David Humphries, Nikki Vaughan and Jue Wang for sharing their design expertise, and to Deborah Ferreira, Mokanarangan Thayaparan and Marco Valentino for sharing their neural network expertise, as part of Sect 7. Thanks also to anonymous reviewers for providing useful feedback on an earlier version of this paper.

## Author contributions

**Conceptualization:** Guy Marshall, André Freitas.

**Data curation:** Guy Marshall.

**Investigation:** Guy Marshall.

**Methodology:** Guy Marshall, Caroline Jay.

**Project administration:** Guy Marshall.

**Supervision:** André Freitas, Caroline Jay.

**Writing – original draft:** Guy Marshall.

**Writing – review & editing:** Guy Marshall, André Freitas, Caroline Jay.

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
