## [Decision Letter · Decision Letter 0]

13 Sep 2024

PONE-D-24-35116A guidelines-based framework for scholarly neural network system diagramsPLOS ONE

Dear Dr. Marshall,

Thank you for submitting your manuscript to PLOS ONE. After careful consideration, we feel that it has merit but does not fully meet PLOS ONE’s publication criteria as it currently stands. Therefore, we invite you to submit a revised version of the manuscript that addresses the points raised during the review process.

We look forward to receiving your revised manuscript.

Kind regards,

Rabie Adel El Arab

Academic Editor

PLOS ONE

Journal Requirements:

2. Thank you for stating the following in the Acknowledgments Section of your manuscript: Guy Marshall acknowledges the support of the Department of Computer Science, University of Manchester. Thanks to David Humphries, Nikki Vaughan and Jue Wang for sharing their design expertise, and to Deborah Ferreira, Mokanarangan Thayaparan and Marco Valentino for sharing their neural network expertise, as part of Section 7. Thanks also to anonymous reviewers for providing useful feedback on an earlier version of this paper.

Please remove any funding-related text from the manuscript and let us know how you would like to update your Funding Statement. Currently, your Funding Statement reads as follows: GM: PhD stipend awarded by University of Manchester Department of Computer Science.

Additional Editor Comments:

Dear Authors,

Thank you for submitting your manuscript, "A Guidelines-Based Framework for Scholarly Neural Network System Diagrams" for review. I have identified a few areas where you might consider/clarify

1. Introduction:

The introduction discusses various broad topics (neural networks, AI research, scholarly communication), which dilutes the central issue. I suggest sharpening the focus on the specific problem of inconsistent diagramming practices in neural network system publications.

Overgeneralization: There is an overgeneralized discussion of the importance of diagrams across various domains, but the introduction does not convincingly argue the specific need for guidelines in neural network systems. Narrowing the problem statement would strengthen the argument.

The introduction lacks a clear hypothesis or research question. It would help readers if you clearly stated the aim of the study early in the introduction.

2. Objectives:

Ensure that the objectives are directly aligned with the problem outlined in the introduction. Currently, they are somewhat broad and lack specific, actionable goals.

3. Methodology:

The use of mixed methods (interviews, card sorting, corpus analysis) is not well integrated, and it’s unclear how the qualitative and quantitative data work together. Clarify how these methods were used in conjunction to arrive at your conclusions.

The fact that participants were already known to the research team could introduce selection bias, and this should be acknowledged in the methodology.

The methodology does not provide enough details on the data analysis process, particularly for the quantitative measures. A more transparent description of how the data was analyzed and interpreted would improve this section.

4. Results:

The results rely heavily on qualitative feedback and lack robust statistical analysis, particularly in demonstrating the efficacy of the framework. Consider strengthening the results by including more quantitative analysis.

: Separate your qualitative and quantitative results more clearly, and ensure that your quantitative findings are presented with adequate statistical support.

5. Discussion:

=The discussion makes broad claims about the framework's effectiveness without sufficient empirical evidence, given the small sample size and limited evaluation. Please revise the discussion to temper conclusions based on the limitations of the data.

The discussion does not critically assess the weaknesses in your study, such as discrepancies between expert and non-expert evaluations and the limitations in diagram interpretation without text. Include a more critical discussion of these issues.

There is little discussion about how this framework can be applied in real-world settings or what barriers exist to its implementation. We encourage you to reflect on the practical implications and challenges of adopting this framework.

6. Conclusion:

The conclusion overstates the impact of the framework. Given the limitations of the study, particularly the small sample size and lack of extensive quantitative support, we suggest tempering the conclusions to match the strength of your evidence.

Provide clear recommendations for future research. Outline how this framework could be tested on a larger scale and how it could be refined for real-world use.

7. Figures and Tables:

There are some inconsistencies between your framework guidelines and the diagrams presented. For example, Figures 6, 7, and 8 include multiple types of arrows, which contradict your guideline about using one type of arrow for information flow. Please ensure that the figures adhere to the guidelines you propose.

Terms like “correctness” and “compliance” in Tables 11 and 12 are not clearly defined, leading to potential confusion. We recommend clarifying these terms and ensuring they align with your textual explanations.

Overinterpretation of Data: In Figure 10, the use of LOESS smoothing could overfit the relationship between compliance and citation counts. Consider adding disclaimers about the limitations of statistical models used in these figures.

8. Self-Citations and Old References:

Given the rapid pace of developments in AI and neural networks, we suggest updating your literature review with more recent works (from the last 5–10 years) that are relevant to the topic.

Looking forward for the revised version

Best regards,

Reviewers' comments:

Reviewer's Responses to Questions

**Comments to the Author**

1. Is the manuscript technically sound, and do the data support the conclusions?

Reviewer #1: Partly

Reviewer #2: Yes

2. Has the statistical analysis been performed appropriately and rigorously? 

Reviewer #1: Yes

Reviewer #2: Yes

3. Have the authors made all data underlying the findings in their manuscript fully available?

Reviewer #1: Yes

Reviewer #2: Yes

4. Is the manuscript presented in an intelligible fashion and written in standard English?

Reviewer #1: Yes

Reviewer #2: Yes

5. Review Comments to the Author

Reviewer #1: Pros:

1. The research question addressed is highly relevant to the neural network community. The inclusion of the diagram is particularly helpful for understanding the proposed framework.

2. The paper presents a range of foundational knowledge, including but not limited to the use of diagrams in communicating neural network systems and references to key publications in the field. This aids in establishing the necessary background for readers.

3. The study includes a comprehensive set of experiments, accompanied by well-structured conclusions and discussions.

Cons:

1. Could the authors clarify why the study predominantly focuses on CV and NLP topics, given that only one participant out of 12 in Table 1 is from an AI or science background? Since the paper addresses neural network systems, further exploration and discussion of other NN-related domains beyond CV and NLP would be beneficial.

2. In Chapter 4, the authors rely heavily on statements from 12 participants. I question whether this method is entirely appropriate, as the responses may contain excessive subjective bias. A more in-depth discussion of the validity of this experimental approach, along with supporting citations, would strengthen the paper.

3. The first-line indentation is inconsistent throughout the paper; some sections have indentation, while others do not.

4. There are formatting issues with the references, including but not limited to L164, L217, and L228.

Reviewer #2: This paper studies the diverse uses and understandings of scholarly neural network system diagrams, proposing an improvement framework based on existing design, information visualization, and user experience principles. Through a combination of interviews, card sorting, and qualitative feedback, the research reveals the diversity and individual preferences in creating and interpreting these diagrams. Additionally, the paper evaluates the effectiveness of the framework through a mixed-methods experimental study and a "corpus study" of published diagrams, aimed at enhancing the communicative efficiency of scholarly neural network diagrams.

Paper Strengths

1.Innovative research methodology: The article employs a mixed-methods research design that combines qualitative and quantitative approaches to provide a comprehensive analysis of the use of neural network system diagrams in academia.

2.Integration of theory and practice: By using ecologically-derived examples combined with theoretical analysis, the study systematically improves existing neural network system diagrams, enhancing the efficiency of information transmission in diagrams.

3.Thorough literature review: The article provides a solid theoretical foundation for its research design and results analysis by thoroughly reviewing relevant studies in the field.

Paper Weaknesses

1.Insufficient experimental details: The paper lacks detailed descriptions of the experimental setup and parameter adjustments, which may affect the reproducibility of the results.

2.Need for clearer exposition: Some sections of the paper, especially in methodology and results interpretation, are not clearly articulated, which may hinder readers' understanding.

Questions to Authors and Suggestions for Rebuttal

1.Could the authors provide more details on the experimental design, particularly the specific steps of data collection and analysis?

2.Given the diversity of diagram designs, how do the authors ensure the general applicability of the proposed framework?

3.Do the authors plan to further expand this study, for example, by incorporating more baseline comparisons or broader field applications?

Overall,this paper proposes a valuable research framework, offering improvements for the design and interpretation of neural network system diagrams in academia. Despite some issues with experimental details and clarity of exposition, the methodological approach and comprehensive review of existing literature enhance its academic value. Based on the above, it is recommended that the paper be accepted after revisions.

6. PLOS authors have the option to publish the peer review history of their article (what does this mean?). If published, this will include your full peer review and any attached files.

Reviewer #1: No

Reviewer #2: No

---

## [Author Response · Author response to Decision Letter 1]

5 Nov 2024

The paper has undergone substantial changes based on the feedback, with particular attention given to precise description of methodology and to limitations, together with the framing of the study. Formatting changes have also been done to the figure formats, referencing, and text.

---

## [Decision Letter · Decision Letter 1]

22 Jan 2025

An evidence-based guidance framework for neural network system diagrams

PONE-D-24-35116R1

Dear Dr. Marshall,

We’re pleased to inform you that your manuscript has been judged scientifically suitable for publication and will be formally accepted for publication once it meets all outstanding technical requirements.

Kind regards,

Shahid Nazir Bhatti, PhD

Academic Editor

PLOS ONE

Additional Editor Comments (optional):

We are pleased to inform you that your manuscript, titled "[An evidence-based guidance framework for neural network system diagrams]", tentatively meets the thematic and quality standards of [PLOS One]. Based on the reviewers’ feedback and our evaluation, your article satisfies the minimum requirements for publication.

Please address the reviewers' comments and suggested revisions to ensure final acceptance if anything pending in this. Once the revisions are complete, we will proceed with the next steps.

Thank you for your valuable contribution.

Best regards,

[Prof. Dr. Shahid]

Editor, [PLOS One]

Reviewers' comments:

Reviewer's Responses to Questions

**Comments to the Author**

1. If the authors have adequately addressed your comments raised in a previous round of review and you feel that this manuscript is now acceptable for publication, you may indicate that here to bypass the “Comments to the Author” section, enter your conflict of interest statement in the “Confidential to Editor” section, and submit your "Accept" recommendation.

Reviewer #1: All comments have been addressed

Reviewer #2: All comments have been addressed

2. Is the manuscript technically sound, and do the data support the conclusions?

Reviewer #1: Yes

Reviewer #2: Yes

3. Has the statistical analysis been performed appropriately and rigorously? 

Reviewer #1: Yes

Reviewer #2: Yes

4. Have the authors made all data underlying the findings in their manuscript fully available?

Reviewer #1: Yes

Reviewer #2: Yes

5. Is the manuscript presented in an intelligible fashion and written in standard English?

Reviewer #1: Yes

Reviewer #2: Yes

6. Review Comments to the Author

Reviewer #1: Thank you for author's response as well as the refined manuscript. My concerns are fully addressed. So I would like to recommend accept this manuscript.

Reviewer #2: During the review process, I carefully read and assessed the content of the article and the revisions made based on the feedback from the initial review. Overall, this article is based on rigorous research methods and proposes an innovative framework for neural network system diagrams, which has been evaluated through experimental research and an extensive literature review. The practicality and usability of the framework have been validated, and it meets the academic standards and expectations of this field.

Therefore, based on the above reasons, I recommend the acceptance of this article for publication. The article provides valuable perspectives and tools for the neural network research community in the representation of system diagrams, which will contribute to the academic development and deeper research in this field.

7. PLOS authors have the option to publish the peer review history of their article (what does this mean?). If published, this will include your full peer review and any attached files.

Reviewer #1: No

Reviewer #2: No

---

## [Editor Report · Acceptance letter]

PONE-D-24-35116R1

PLOS ONE

Dear Dr. Marshall,

I'm pleased to inform you that your manuscript has been deemed suitable for publication in PLOS ONE. Congratulations! Your manuscript is now being handed over to our production team.

Kind regards,

on behalf of

Dr. Shahid Nazir Bhatti

Academic Editor

PLOS ONE